# Ethnomedicinal Plants and Herbal Preparations Used by Rural Communities in Tehsil Hajira (Poonch District of Azad Kashmir, Pakistan)

**DOI:** 10.3390/plants13101379

**Published:** 2024-05-15

**Authors:** Tahira Jabeen, Muhammad Shoaib Amjad, Khalid Ahmad, Rainer W. Bussmann, Huma Qureshi, Ivana Vitasović-Kosić

**Affiliations:** 1Department of Botany, Women University of Azad Jammu & Kashmir Bagh, Bagh 12500, Pakistan; tj969260@gmail.com; 2Department of Environmental Sciences, COMSATS University Islamabad, Abbottabad Campus, Abbottabad 22060, Pakistan; khalidahmad@cuiatd.edu.pk; 3Department of Ethnobotany, Institute of Botany, Ilia State University, Tbilisi 0162, Georgia; rainer.bussmann@deiliauni.edu.ge; 4Department of Botany, State Museum of Natural History, 76135 Karlsruhe, Germany; 5Department of Botany, University of Chakwal, Chakwal 48800, Pakistan; huma.qureshi@uoc.edu.pk; 6Division of Horticulture and Landscape Architecture, Department of Agricultural Botany, University of Zagreb Faculty of Agriculture, Svetošimunska cesta 25, 10000 Zagreb, Croatia

**Keywords:** traditional knowledge, ethnobotanical survey, medicinal, ethnopharmacology herbal tradition, Pakistan

## Abstract

The present study emphasizes the importance of documenting ethnomedicinal plants and herbal practices of the local rural communities of Tehsil Hajira (Pakistan). The aim was to document, explore and quantify the traditional ethnomedicinal knowledge. Ethnobotanical data were collected using semi-structured questionnaires and analyzed using various quantitative indices. The results showed that 144 medicinal plant species from 70 families and 128 genera play an important role in herbal preparations. The most common type of preparation was powder (19.0%), followed by paste (16.7%), aqueous extract (15.7%), decoction (14.7%) and juice (11.0%). *Fragaria nubicola* (0.94) and *Viola canescens* (0.93) had the highest relative frequency of mention (RFC), while *Berberis lycium* (1.22) and *Fragaria nubicola* (1.18) had the highest use value (UV). *Geranium wallichianum* (85.5), *Ligustrum lucidum* (83) and *Indigofera heterantha* (71.5) were the most important species in the study area with the highest relative importance (RI) value. The diseases treated were categorized into 17 classes, with diseases of the digestive system and liver having the highest Informant Consensus Factor (ICF) value, followed by diseases of the oropharynx and musculoskeletal system. Important plants mentioned for the treatment of various diseases of the gastrointestinal tract are *Zanthoxylum alatum*, *Berberis lycium*, *Mentha longifolia*, *Punica granatum*, *Rubus ellipticus* and *Viola canescens*. New applications of rarely documented plants from this area are: *Oxalis corniculata* paste of the whole plant to treat vitiligo, *Carthamus tinctorius* flowers to treat chicken pox, *Dioscorea deltoidea* tuber powder to treat productive cough, *Inula cappa* root decoction to treat miscarriage, *Habenaria digitata* tuber juice for the treatment of fever, *Viola canescens* leaves and flowers for the treatment of sore throat and *Achillea millefolium* root and leaf juice for the treatment of pneumonia. These plants may contain interesting biochemical compounds and should be subjected to further pharmacological studies to develop new drugs. Traditional medicinal knowledge in the area under study is mainly limited to the elderly, traditional healers and midwives. Therefore, resource conservation strategies and future pharmacological studies are strongly recommended.

## 1. Introduction

Traditional medicine (TM) is based on the knowledge and beliefs of local communities and is considered the oldest form of medical practice [1,2]. The ingredients used in TM are usually naturally occurring plants, animals and minerals [3]. About 60% of the world’s population and 80% of the population in developing countries rely on traditional medicine [4], mainly because of its ease of access [5,6], its belief and trust [7,8], its perceived safety and efficacy [9] and its inaccessibility to modern medicine. One-third of the world’s population has no regular access to modern primary health care [10], while half of the population in Africa, Asia and Latin America faces a lack of minimal medical care [11], although in many areas a coexistence of conventional and traditional medicine has been observed [12].

Of the approximately 295,000 flowering plants in the world, less than 10% have been studied for their medicinal properties [13]. There is indeed a growing interest in researching medicinal plants for the development of allopathic medicines, and this trend is driven by various factors such as the search for new sources of medicines, the need for more sustainable and natural alternatives and the recognition of traditional systems of medicine [14,15,16]. Herbal products and their derivatives still account for about 25–50% of pharmaceuticals worldwide [17,18,19]. A variety of chemical constituents are extracted from plants, which are valuable resources for the pharmaceutical, cosmetics and food industries. Alkaloids such as morphine, quinine and nicotine, terpenes found in essential oils of plants such as lavender, citrus and cannabis, and flavonoids abundant in fruits such as berries (anthocyanins), citrus fruits and vegetables such as onions (quercetin) serve a variety of purposes [20]. The ethnomedicinal use of plants should be documented in a standardized format [21,22,23,24]. Standardized documentation ensures consistency and facilitates the comparison and analysis of traditional knowledge in different cultures and regions. This approach records detailed information about the plant species, parts used, preparation methods, dosage, therapeutic use and associated cultural practices or beliefs. Such standardized documentation not only preserves valuable traditional knowledge but also makes it more accessible for scientific research, nature conservation and the sustainable use of plant resources. Understanding the properties of raw materials is important to preserve a national heritage [25,26], but the understanding of traditional medicine is still limited, even in Asian countries [27,28].

Since the Alma Ata Declaration (late 1970s), the WHO has reaffirmed its commitment to promote the development of public policies to incorporate traditional practices into the health systems of all Member States [29]. The inhabitants of developing countries are the most frequent users of these practices, and 67% of the world’s plant species are found in these countries [30]. Herbal medicines are a cost-effective source of primary health care, especially where modern health facilities are lacking [31,32]. Efforts have been made to integrate herbal preparations into modern medical systems to improve access to health care [33,34,35,36], and patients often take traditional and allopathic medicines simultaneously [12]. Unfortunately, plant resources are often under significant anthropogenic stress, leading to an alarming decline in plant diversity and associated traditional knowledge. Other factors contributing to the loss of traditional knowledge are modern schooling, land use changes, the market economy, and the process of industrialization and globalization [37].

In the mountainous regions of Azad Kashmir in northern Pakistan, there is a great diversity of plant habitats, soil types and climatic conditions. Many endemic plants of Pakistan are restricted to this region. There are increasing reports of local societies in Azad Jammu and Kashmir (AJK) using traditional medicinal plants [38,39,40], but comprehensive documentation of local useful flora and associated traditional knowledge is still lacking. Local traditional healers and herbalists play an important role as they provide health services to about 75% of the rural population [41,42]. The Poonch Valley, an administrative division of Azad Jammu and Kashmir, is particularly rich in biodiversity but hardly explored for ethnobotanical knowledge [43,44]. Many remote areas like Tehsil Hajira are still unexplored. We hypothesized that ethnomedicinal knowledge varies widely across different regions of Pakistan and that there are some novel uses of plants, especially in remote areas like Tehsil Hajira. This hypothesis is based on the assumption that geographical isolation and different cultural practices may lead to unique ethnomedicinal traditions within specific communities. Therefore, we hypothesize that the ethnomedicinal knowledge documented in the Tehsil Hajira area would show remarkable differences compared to other regions of Pakistan. The aim of the current study was to (a) document the traditional knowledge of medicinal plants used to cure various diseases, (b) compare the documented ethnomedicinal knowledge using quantitative indices, specifically use value, relative frequency of mention, degree of fidelity, and informant approval factor, and (c) identify new medicinal plants and novel uses by comparing the present findings with previously reported uses from neighboring areas.

## 2. Results and Discussion

### 2.1. Demographic Data of the Informants

A total of 70 informants (40 men and 30 women) were interviewed as part of the present study (Table 1). Mainly due to the traditional concept of gender segregation, honor and covering (parda) in the study area, the proportion of female respondents was 25% less than that of males. Furthermore, women are not allowed to communicate with strangers. Nevertheless, female respondents in the study area had more knowledge (they named an average of 6.22 species) than male respondents (who named an average of 5.56 species). This reflects their important contribution to household management and maintaining the health of the family. Similar trends have been observed in other parts of the country such as the Chail Valley [45], Balochistan [46] and Toli Peer [47] as well as globally, e.g., in Brazil [48].

The informants were categorized into three age groups, namely 20–40 years, 41–60 years and 61–80 years old. The older informants were more knowledgeable (they named an average of 12.10 species with 10.90 uses), followed by the middle-aged participants (8.96 species with 5.40 uses) and the younger ones (4.22 species with 3.56 uses; Table 1), suggesting that the older ones had more knowledge that they had acquired during their lifetime. Nowadays, the social fabric is changing due to socioeconomic changes and advances in science and technology, even though the study area is remote and close to the Line of Control between India and Pakistan, where mobility is restricted due to the tense security situation. Nevertheless, the young generation is no longer very interested in learning about this traditional wealth. This phenomenon was frequently observed, both on the Asian continent and in various cultural contexts worldwide [45,46,49,50].

Most of the informants had a very low level of education, and only seven informants had a higher education. The level of education was inversely related to traditional knowledge, as uneducated natives were generally more familiar with the use of ethnomedicinal plants than educated individuals (Table 1). The reason for this appears to be a lack of preference for learning and applying ethnobotanical knowledge as part of “modern education”. The same findings have been documented by other researchers in other areas of Pakistan [49,50,51] and abroad [52,53].

### 2.2. Traditional Health Systems

Local health systems make an important contribution to primary health care worldwide, especially in Asian and African countries. Many people in Pakistan and Azad Kashmir rely on traditional remedies to cure various diseases, especially in marginalized communities [54]. The present data (Table 1) show that among different occupational groups, ethnomedicinal knowledge was highest among traditional health practitioners (mean 23.55 species; 12.4 uses) and midwives (14.2 species; 9.83 uses). In the region, traditional health practitioners (THPs) were often old men and had the most information on the medicinal use of plants, minerals and animals to cure chronic diseases, while midwives were experienced older women who were familiar with pregnancy problems and treated them with herbal medicines. These details have also been described elsewhere in the literature [55,56,57,58]. Pastoralists also had significant ethnomedicinal information (9.18 species; 8.62 uses), as nomadic pastoralist communities not only have direct experience with the use of plants but are also important collectors of medicinal plants, especially at higher altitudes, as also reported from Mustang, Nepal [59]. However, as reported earlier, traditional knowledge of herbal medicine is steadily declining due to better availability of modern health facilities and changing lifestyles [54,60,61], a trend also observed in the study area.

### 2.3. The Diversity of Ethnoflora

In the present study, 144 species from 70 families and 128 genera were found (Table 2). Asteraceae was the dominant family (16 species), followed by Rosaceae (11 species), Lamiaceae (8 species) and Fabaceae (8 species) (Figure 1). The dominance of these families in the study area could be due to the suitability of the habitat and environment for the members of these families. Similar results were reported in Brazil, Ethiopia and India [47,51,62,63,64]. The frequent use of these families could also be due to easy access, knowledge of long-term use and high content of bioactive constituents. Most of the herbal preparations in the study area were based on herbaceous species followed by shrubs and trees (Figure 2). The probable reason for this is the abundant growth and easy availability of herbaceous species, while the lesser number of trees could be due to climatic and environmental factors such as altitude, etc. These findings are in line with other studies from Pakistan [65,66,67,68,69,70,71]. Indeed, land use practices such as deforestation and the conversion of natural habitats to intensive agriculture can contribute significantly to genetic erosion, especially in regions such as Pakistan. These factors can lead to a loss of biodiversity, both in herbaceous and tree species, affecting the availability and abundance of medicinal plants. In addition, changing land use patterns can disrupt ecosystems and affect the distribution and survival of plant species. Therefore, although altitude and other environmental factors play a role, it is important to consider the broader effects of land use change on genetic erosion and the diversity of medicinal plants in the region.

### 2.4. Mode of Administration and Plant Part(s) Used in Herbal Preparations

The most common type of preparation was powder (19.0%), followed by paste (16.7%) in which the plant material is usually mixed with water, aqueous extract (15.7%), decoction (14.7%) and juice (11.0%) (Figure 3). The frequent use of powder could be due to its ease of preparation and efficacy in herbal medicines as reported from Pakistan and other countries [46,49,79,86,87]. However, it should be noted that powder is not always the easiest method of using herbal remedies. In the area of internal use, for example, infusions can offer a simpler approach. Other well-known preparation methods include decoctions, infusions, herbal teas and decoctions. “Herbal tea” refers to a beverage made by briefly steeping dried herbs, flowers, fruits or other botanicals in hot water, extracting flavors and some medicinal compounds, while “infusion” encompasses a broader range of beverages made by steeping botanicals in hot water, including herbal teas as well as leaves, fruits or spices. Decoction, on the other hand, “involves the boiling of harder plant parts such as roots or seeds in water for a prolonged period of time, often at low heat, to release medicinal compounds,” while “cooking” is a general cooking technique in which water is heated to boiling point for various culinary purposes, including the preparation of infusions or herbal teas, but without the particular emphasis on prolonged boiling required for decoctions. The most commonly used plant parts were leaves (28.8%), followed by roots (13.5%), whole plants (12.8%) and fruits (11.5%) (Figure 4). A large number of bioactive constituents stored in the various parts of the plant could be beneficial for local residents in the treatment of various ailments [82,88,89]. Certain classes of bioactive compounds such as alkaloids, flavonoids and terpenes are often favored in traditional medicine for their therapeutic properties. However, further analysis would be required to identify specific trends in the use of these ingredients. The leaves are particularly abundant and more easily accessible than other parts of the plant such as seeds and roots. Another reason for collecting leaves is their higher sustainability compared to other parts [47]. Leaves have been reported to be used by many different communities worldwide for the preparation of various herbal products [47,49,90,91,92,93,94]. For example, crushed or ground leaves of plants such as *Aloe vera*, eucalyptus and neem are used to make poultices to treat skin diseases, wounds and insect bites. In addition, the leaves of culinary herbs such as basil, coriander, parsley and bay leaf are often used as flavorings in the kitchen to enhance the taste and aroma of food. The fact that roots are the second most used plant part in herbal medicine after leaves can be attributed to their rich concentration of bioactive compounds. Roots serve as a reservoir for various phytochemicals, including alkaloids, flavonoids, terpenoids and phenolic compounds, which exhibit various pharmacological activities such as anti-inflammatory, antioxidant, antimicrobial and anticancer properties. In addition, specialized structures within the roots, such as rhizomes or tubers, store energy reserves and protective substances, which further increases their medicinal potential. The physiological functions of roots, including water and nutrient uptake, can also influence their biochemical composition, making them valuable resources for traditional healers and herbalists seeking effective herbal remedies.

### 2.5. Relative Frequency of Citation (RFC)

The RFC defines the most frequently occurring plant species in various diseases used by the locals. In our study, the RFC ranged from 0.02 to 0.94. *Fragaria nubicola* (0.94), *Viola canescens* (0.92) and *Mentha longifolia* (0.87) had the highest RFC values, while *Cyperus rotundus* and *Bidens bipinnata* (0.02) had the lowest values (Appendix A Table A1). A high RFC value indicates that the availability and knowledge of a particular plant species is very high. Such species may be of interest for marketing, pharmacological and phytochemical studies to assess and prove their validity [60,95]. These species should be protected as a priority, as their overexploitation can endanger the local populations. Many species with maximum RFC are collected by healers and cultivated in gardens not only for ethnomedicinal purposes but also for ornamental and conservation purposes. The present results are consistent with other studies [47,50].

### 2.6. Utilization Value (UV)

The use value is the quantity of uses recognized for a particular plant species. It indicates the extent to which a species is used in a particular area. Species with a high UV are well known to informants and are extensively used in ethnomedicinal preparations [49,89], probably because they contain more biologically active compounds [96]. Indeed, certain classes of compounds are often associated with high UV, such as alkaloids, flavonoids and phenolic compounds. These bioactive compounds are known for their medicinal properties and are often used in ethnomedicinal practice for their potential therapeutic effects. In the current study, *Berberis lycium* (1.22), *Fragaria nubicola* (1.18) and *Viola canescens* (1.07) had the highest use values (Appendix A Table A1). However, plants with low RFC and UV values are not insignificant, but young people are less aware of their use, and knowledge may eventually disappear, as in other areas [97]. We compared the documented plant uses with previously published studies from other areas (Appendix A Table A1) [42,60,84,98]. Similar to Bano et al. [60], *Hippophae rhamnoides* and *Rosa brunonii* had the highest values, ranging from 1.64 to 1.47. The correlation analysis showed a strong positive correlation between UV and RFC at a significant level of 0.01 (Figure 5). This can be attributed to the fact that the plants that are well known in the local communities are extensively used in herbal preparations. Thus, if a plant is well known in the communities, it has a higher use value and is important for pharmacological evaluation.

### 2.7. Relative Importance (RI)

The species with high values for relative importance are often versatile. The highest RI values were found for *Geranium wallichianum* (85.5), *Ligustrum lucidum* (83) and *Indigofera heterantha* (71.5) (Table 3). Plants with a high RI showed a higher number of uses in the treatment of various body systems, and local people passed on a wealth of information about these plant species. The importance of plant species is increasing due to their ability to cure diseases [99]. This suggests that these plants are highly valued in traditional medicine, possibly due to their purported efficacy in treating ailments in various body systems. As a result, the importance of these plant species is increasing, especially among the general population, where their potential to cure diseases is increasingly recognized.

### 2.8. Informant Consensus Factor (ICF)

The ICF values ranged from 0.66 to 0.91 (Figure 6). All diseases were categorized into 17 different types. The highest ICF score was found for digestive disorders (0.91). Due to the insufficient availability of clean drinking water and hygienic food, digestive disorders are widespread in this area [100,101]. Important plants in this category were *Zanthoxylum alatum, Berberis lycium, Mentha longifolia, Punica granatum, Rubus ellipticus* and *Viola canescens* (Figure 6). Important plants for curing various gastrointestinal ailments have been documented for different ethnic communities around the world [102,103]. These plants contain active ingredients that are recognized by the local population and are used to treat various gastrointestinal disorders. Other studies have also found a high consensus factor value for gastrointestinal complaints [46,104,105,106,107,108]. The local population was very familiar with these species, which were used to treat a variety of ailments. The lowest ICF value (0.66) was found for hair care.

### 2.9. Fidelity Level (FL)

The Fidelity Level (FL) of the 69 most important plant species ranged from 12.5% to 100%. A high FL value generally indicates the dominance of a particular ailment in a region and the use of the same species to treat it [46,109]. The species with an FL value of 100 were *Berberis lycium, Fragaria nubicola, Punica granatum, Viola canescens, Mentha longifolia* and *Elaeagnus umbellata*, which were used for wound healing, mouth infections, jaundice, fever, stomach ailments and toothache (Figure 7). Plant species with a high FL are considered ideal for ethnopharmacological studies [110]. However, plants with a low FL should not be underestimated [111].

### 2.10. Novelty and Future Implications

The present study was compared with 22 different ethnobotanical studies from other parts of Azad Kashmir and the rest of Pakistan using the Jaccard index. The value of the Jaccard index ranged from 1.02 to 22.27. The highest degree of similarity was found for nearby areas like Pearl Valley [44] and Toli Peer [47] (Table 3).

This higher degree of similarity could be due to similar climatic conditions, vegetation types and frequent intercultural exchanges between these ethnic communities. The lowest similarity with other studies could be the result of extensive changes in population and habitat structure, showing a high degree of cultural adaptation [112]. Any change in these parameters would lead to a change in the similarity index values, as these indices are very sensitive to changes in population size [72].
plants-13-01379-t003_Table 3Table 3Jaccard index comparing the present study in Tehsil Hajira with previous published studies.Study AreaSYNRPsNPSUNPDUTSCBASEAASESAPPSUPPDUJIC**From Azad Jammu and Kashmir (AJK), Pakistan**Poonch Valley 20126842327411175.8833.814.59[72]Leepa Valley 20123621113231315.5530.557.78[74]Pearl Valley 2017136371451859327.210.2922.27[44]Toli Peer 2017121301545769924.712.320.45[47]Kotli2017128202545839915.619.519.82[23]Neelum Valley20175021820301244.0036.0011.49[21]Kotli 20198012284040104153521.74[82]**From Khyber Pakhtunkhwa (KPK), Pakistan**Dir Valley, KPK, 20116532326391184.6135.314.21[113]Chail Valley, Swat2014501582327121301613.47[45]Kabal, Swat20154521719261254.4437.711.17[114]Alpine and Sub-alpine regions of Pakistan201512577141111305.65.65.49[49]Manoor Valley, KPK20164411314301302.2729.548.05[80]Northern Pakistani Afghan borders20189212627651171.0918.0612.91[85]Mohmand Agency, FATA20186411213511311.568.336.67[31]**From other areas of Pakistan**Mastung District, Baluchistan201410211112901320.9810.75.13[46]Turmic Valley, Gilgit-Baltistan201542123391412.384.761.63[115]Hafizabad, Punjab, 20178561016691287.0511.77.51[79]Wazirabad, Punjab, 201831178231363.2222.54.79[81]Chenab riverine area, Punjab Pakistan2019129911201091246.978.527.91[83]**From other parts of the world**Blue Nile State, Sudan2011530225114203.771.02[77]Alasehir (Manisa), Turkey201313709912813506.563.30[76]Jhalawar, Rajasthan, India2015190221742010.51.24[78]**Legend**: SY: Study year, NRPs: Number of reported plant species, NPSU: Number of plants with similar uses, NPDU: Number of plants with different uses, TSCBA: Total species common in both areas, SEAA: Species found in aligned areas, SESA: Species found only in study area, PPSU: Percentage of plant with similar uses, PPDU: Percentage of plant with different uses, JI: Jaccard index, C: citation.


The comparative analysis has brought to light some new uses of plant species rarely documented in this region, such as the use of a paste made from the whole plant of *Oxalis corniculata* to treat vitiligo, the flowers of *Carthamus tinctorius* to treat chicken pox, the tuber powder of *Dioscorea deltoidea* to treat productive cough, the root decoction of *Inula cappa* for the treatment of miscarriages, the tuber juice of *Habenaria digitata* for the treatment of fever, the leaves and flowers of *Viola canescens* for the treatment of sore throat and the root and leaf juice of *Achillea millefolium* for the treatment of pneumonia. These plants may contain interesting bio-chemical compounds and should be subjected to further pharmacological studies to develop new drugs. However, it is important not to forget efficacy and toxicity studies, among others, to ensure the safety and effectiveness of potential pharmaceuticals derived from these plants.

## 3. Materials and Methods

### 3.1. Study Area

Tehsil Hajira is a sub-district of Poonch District in Azad Kashmir in Pakistan and lies at an altitude between 650 and 1950 meters. It is located between longitude 73°53′45.96″ E and latitude 33°37′18.12″ N (Figure 8). The study area includes steep slopes, high mountains and dissected small terraces. The region is located 130 km from Muzaffarabad (the capital of Azad Kashmir) and 160 km from Islamabad (the capital of Pakistan). The total area of Hajira, Azad Jammu and Kashmir is 970 km^2^, and the population is 140,000 according to the 2017 census. The sampling sites were selected based on altitude, vegetation heterogeneity and physiognomy. The climate in the region varies from subtropical to temperate, with an average monthly rainfall of 66 mm. The highest amount of precipitation falls in August (114 mm). Temperatures vary between 3.2 °C and 35.6 °C, with June and July being the warmest months with average temperatures of 28.9 °C and 27.6 °C, respectively, while the coldest months are January and December with average temperatures of 8.6 °C and 10.2 °C, respectively (Figure 9). The vegetation is mainly dominated by *Pinus roxburghaii* (Chir), *Pinus wallichiana* (Biar) and *Quercus incana* (Iriana). The ground flora is dominated by various angiosperms as well as mosses and ferns.

Tehsil Hajira has a diverse and complex ethnic composition, including Suddhans, Pathans, Awans, Malik Khwajas, Janjua Rajputs, Dullies, Jaats, Gujars, Ghakhars, Mughals and Qureshis. Most of the people have migrated from Jammu and Kashmir. Among them, the Suddhans and Rajputs are the most influential tribes. The entire population is Muslim. The majority of the people speak Pahari, Hindko and Gujari, and many are also familiar with Urdu. The area has a rural culture with old traditions and its own principles of village life, home, family, festivals and ceremonies. Most of the inhabitants live in poor socio-economic conditions with limited sources of income. Most are farmers, some are employees and laborers, and only a few have their own small businesses. Few public dispensaries and only one small tehsil main hospital (Khai Gata-Hajira Rd., Hajira) provide primary health care, but people living in higher altitudes and remote places have very limited access to these facilities and therefore mainly use herbal remedies for primary health care.

### 3.2. Ethnobotanical Data Collection

The field studies were conducted from March 2017 to September 2017. Data collection involved several visits to each site to ensure a comprehensive understanding and clarification of the information gathered. The ethnobotanical information on the use of plants was collected from a total of 70 informants (40 men and 30 women) through semi-structured interviews and focus group discussions (Table 1). Participants were selected randomly [35] or, in some cases, by snowballing [32]. First, key people with knowledge of the community were identified, and then, as part of a snowballing process, further informants were recommended by these people, thereby expanding the network. Ethical clearance to conduct the study was obtained from the ethics committee of Azad Jammu & Kashmir Women’s College, Bagh, prior to the commencement of the surveys. In addition, legal approval was obtained from the members of the local union council. The code of ethics of the International Society of Ethnobiology (http://www.ethnobiology.net/) was followed. Informed consent was obtained verbally from the informants before the interview commenced. Detailed information about the medicinal use of the plant, the method of preparation and the parts used were recorded.

### 3.3. Floristic Inventory and Botanical Identification

During the field survey, the specimens of each plant species were collected, dried, preserved and mounted on standard-sized herbarium sheets. Identification of the plant specimens was conducted with the help of a taxonomist using the Flora of Pakistan (https://www.eflora.com) [116,117]. The correct family name was determined using APG-IV (2016) [118]. The correct and legitimate names were determined using the Plant List 2013 [119]. Further verification of the identified species was carried out in the Herbarium of Medicinal and Aromatic Plants of Pakistan Agriculture Research Council (PARC). The fully identified voucher specimens were submitted to the herbarium of the Botany Department of AJK Women’s University, Bagh, for further utilization.

### 3.4. Data Analysis

The collected primary ethnobotanical data were converted into quantitative data using the following quantitative indices:

#### 3.4.1. Relative Frequency of Citation (RFC)

The relative frequency of citation of the reported species was calculated according to Vijayakumar et al. [120]:$$Relative\;Frequency\;of\;Citation=\frac{FC}{N}$$
FC = number of respondents who reported the use of a particular species;N = total number of respondents.

#### 3.4.2. Use Value (UV)

The use value was determined according to Vijayakumar et al. [120] by applying the following formula:$$Use\;Value=\Sigma \frac{Ui}{N}$$
where Ui is the number of uses mentioned by each informant for a particular species, and N is the total number of informants.

#### 3.4.3. Informant Consensus Factor (ICF)

The informant consensus factor was derived to determine the consensus among informants on the reported remedies for each disease category. It was calculated based on Heinrich et al. [121]:$$\mathrm{Informant}\;\mathrm{Consensus}\;\mathrm{Factor}=\frac{Nur-Nt}{Nur-1}$$
where Nur is the use reports for each disease group and Nt is the number of species cited by all respondents.

#### 3.4.4. Relative Importance (RI)

The relative importance was determined according to Khan [43] with the following formula:Relative importance=(Rel PH+Rel BS)×100/2
PH = pharmacological property of the selected plant;

Rel PH = the relative number of pharmacological properties reported for a selected plant.
$$Rel\;PH=\frac{PH\;of\;aselected\;plant\;species}{Maximum\;PH\;of\;total\;reported\;species}$$
BS = total body systems cured by a given plant species;

Rel BS = relative number of body systems cured by a particular plant species.
$$Rel\;BS=\frac{BS\;of\;selected\;plant\;species}{Maximum\;PH\;of\;total\;species}$$

#### 3.4.5. Jaccard Index (JI)

This index was used here to identify novel uses by comparing the medicinal uses reported in the current study with previously published work from neighboring regions. It was calculated following Gonza et al. [122] as follows:$$\mathrm{Jaccard}\;\mathrm{Index}=\frac{C\times 100}{\left(a+b\right)-c}$$
a = species of our study area;b = species of the neighboring area;c = species occurring in both areas.

#### 3.4.6. Fidelity Level (FL)

Fidelity level reflects the preference of certain species by the local population for the treatment of a certain disease in the studied region. The valuesfor FL were calculated according to Alexiades and Sheldon [123]:$$\mathrm{FL}\;(\%)=N^{p}/N\times 100$$
N^p^ = informants who requested the use of a specific plant species for a specific condition;N = respondents who named the plant species for any ailment.

The maximum FL of reported plant species shows high preference of plant species for curing a specific disease by the local inhabitants in the investigated region.

#### 3.4.7. Pearson Correlation Analysis 

Correlation analysis between Relative Frequency of Citation and use value was made using the Cor function in R v. 4.3.1 Software.

## 4. Conclusions

Medicinal plants are the basis of traditional health practices in Tehsil Hajira in Poonch District, where, despite the availability of modern health facilities, reliance on herbal remedies remains high, especially among the elderly and alternative practitioners. However, this knowledge is in danger of dwindling due to the dwindling interest of the younger generation. The selection of plant species is of immense importance, not only for local communities but also for research and the pharmaceutical and food industries, as it offers the potential for new therapeutic discoveries. However, this biodiversity is threatened by overexploitation, grazing and deforestation, so urgent conservation measures are needed. Concerted efforts are therefore needed to preserve ethnomedicinal knowledge, sustainably manage medicinal plant resources and unlock their full therapeutic potential through future pharmacological studies.

## Figures and Tables

**Figure 1 plants-13-01379-f001:**
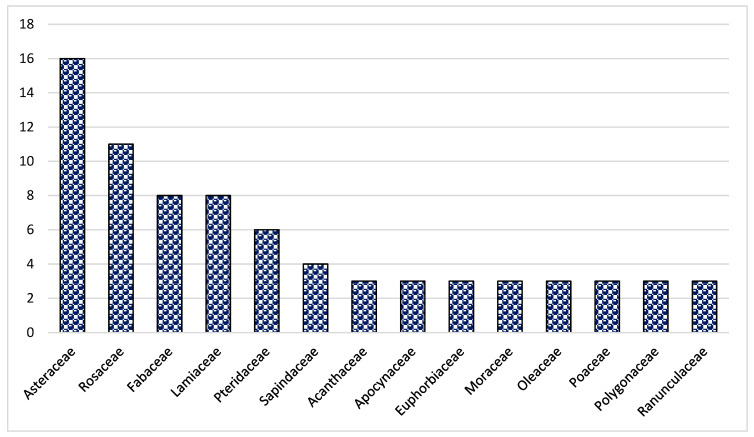
The most abundant medicinal plant families in Tehsil Hajira.

**Figure 2 plants-13-01379-f002:**
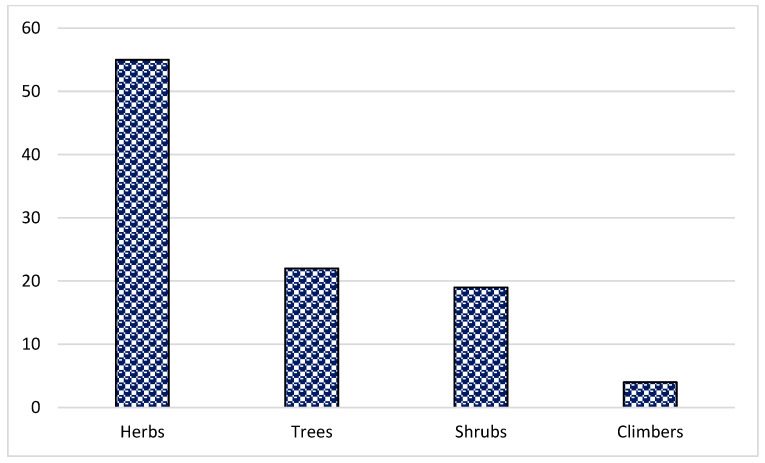
Distribution pattern of life forms of the reported plant species in the study area of Tehsil Hajira (Pakistan).

**Figure 3 plants-13-01379-f003:**
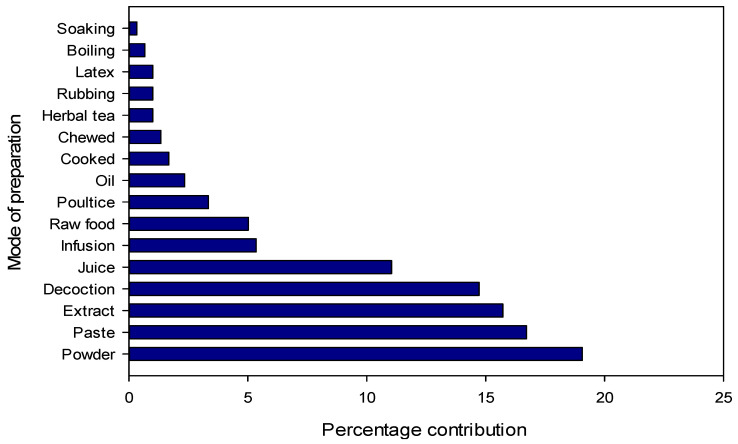
Method of preparation of medicinal plants in Tehsil Hajira (Pakistan).

**Figure 4 plants-13-01379-f004:**
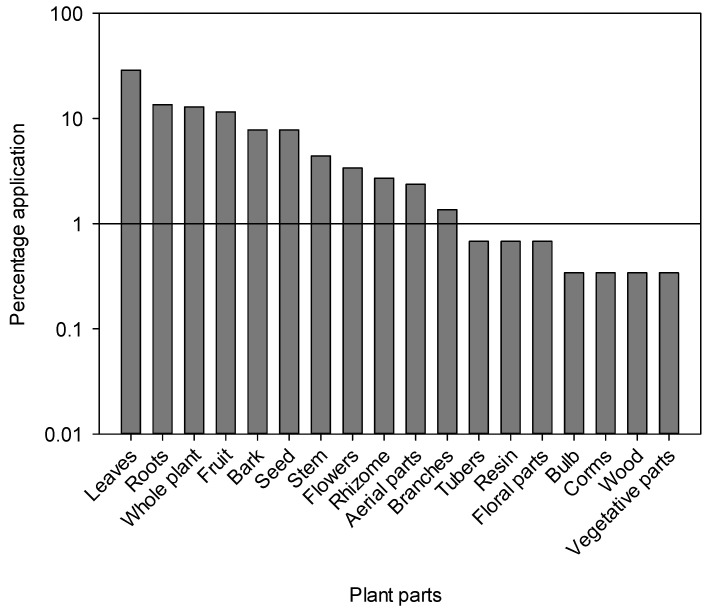
Percentage of use of various plant parts of medicinal plants from the Tehsil Hajira area (Pakistan).

**Figure 5 plants-13-01379-f005:**
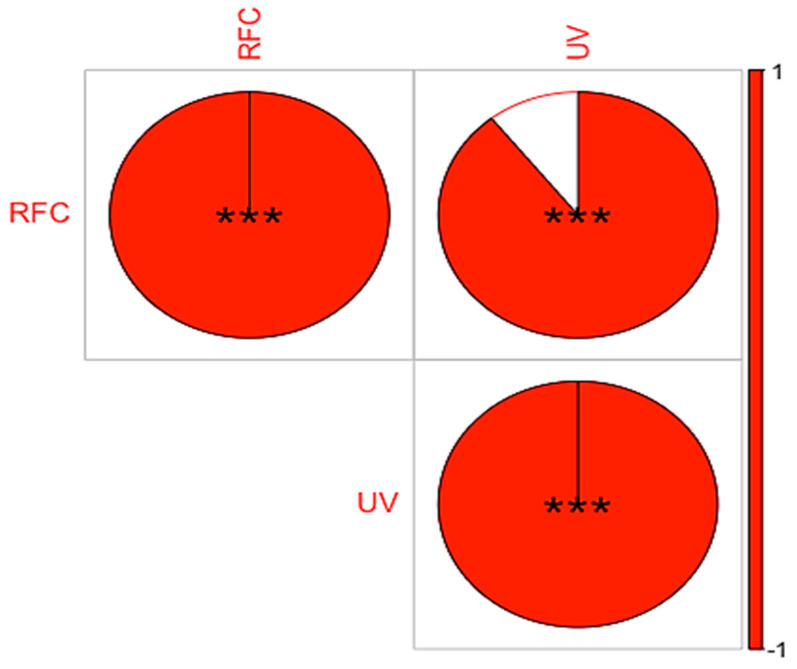
Correlation between RFC and UV of ethnomedicinal plant use in Tehsil Hajira (UV = Use value; RFC = Relative frequency of citation).*** indicate strong positive correlation between UV and RFC at a significant level of 0.01.

**Figure 6 plants-13-01379-f006:**
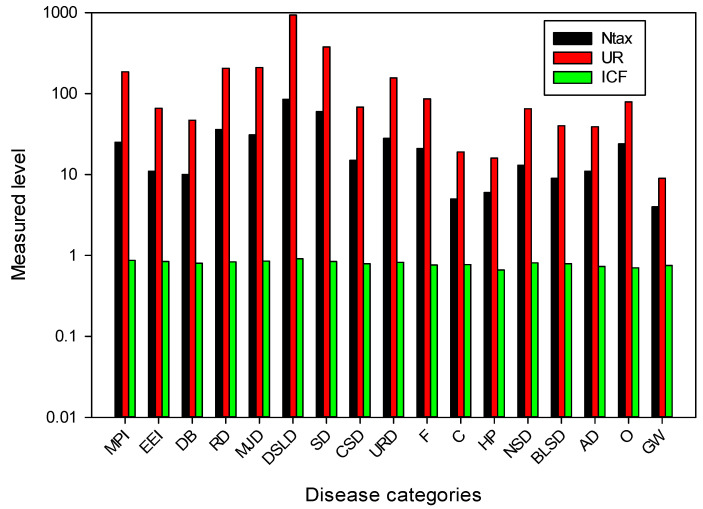
Number of taxa and usage reports of plant species used in the treatment of various diseases in Tehsil Hajira (Pakistan). (Ntax, total number of species used by all informants for a disease group; UR, total number of utilization reports in each disease group; ICF, consensus factor of informants; MPI, mouth and throat infection; EEI, eye and ear infection; DB, diabetes; RD, respiratory diseases; MJD, muscle and joint diseases; DSLD, diseases of the digestive system and liver; SD, skin diseases; CSD, diseases of the circulatory system; URD, diseases of the urinary tract and reproductive organs; F, fever; C, cancer; HP, hair problems; NSD, diseases of the nervous system; BLSD, diseases of the blood and lymphatic system; AD, antidote; O, other; GW, general weakness).

**Figure 7 plants-13-01379-f007:**
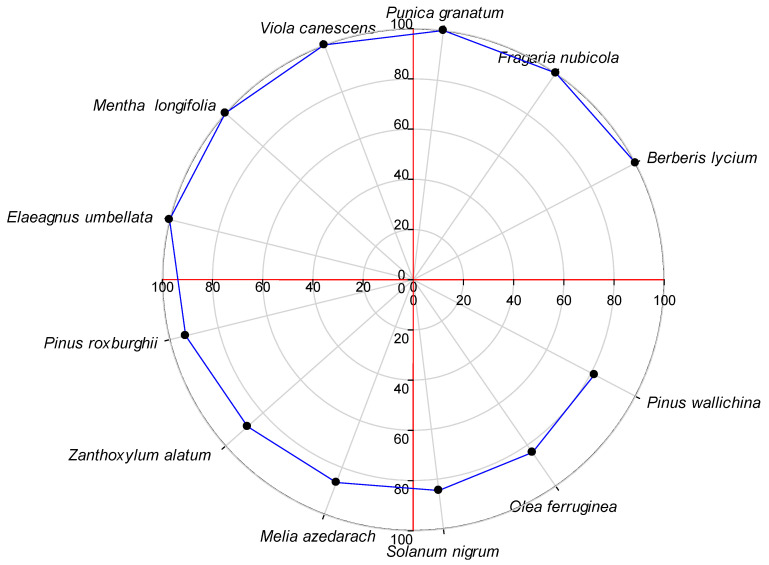
Top ethnomedicinal plant species with a fidelity level of over 80 in Tehsil Hajira.

**Figure 8 plants-13-01379-f008:**
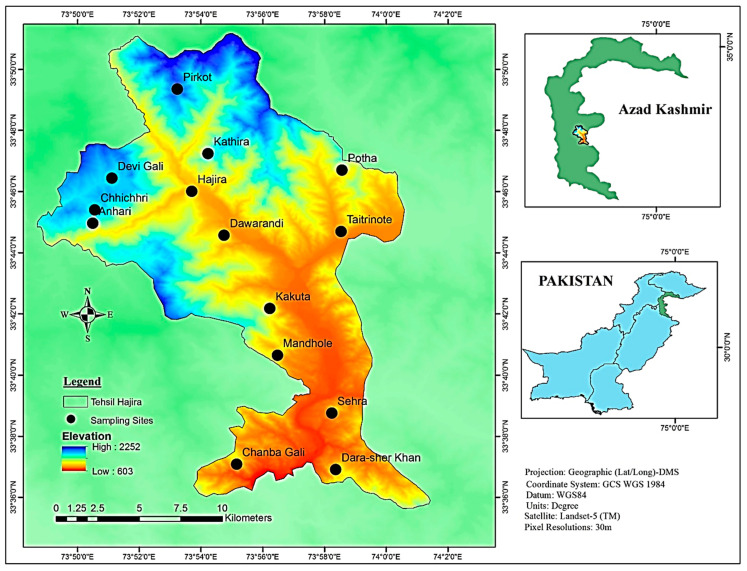
Map of the study area of Tehsil Hajira (Poonch District of Azad Kashmir, Pakistan).

**Figure 9 plants-13-01379-f009:**
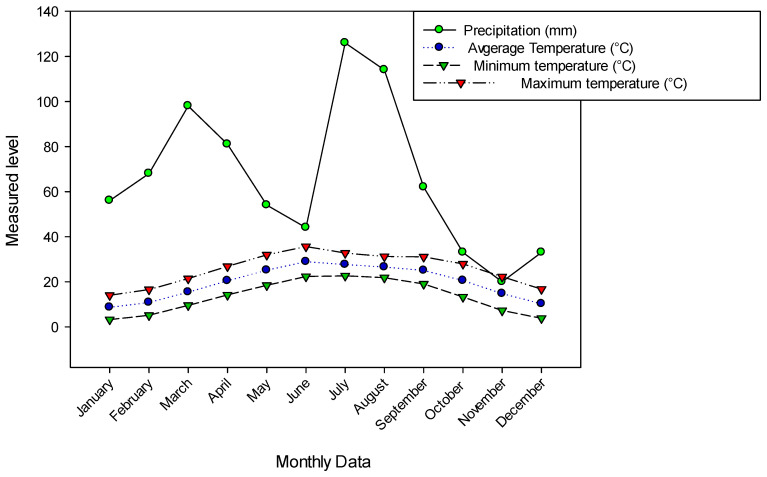
Precipitation and temperature of the study area of Tehsil Hajira.

**Table 1 plants-13-01379-t001:** Demographic data of the informants.

Variables	Informant Category	No of Inf.	ANSRI	ANURI
Gender	Male	40	5.56	8.23
	Female	30	6.22	9.68
Age-Class	20–40	23	4.22	3.56
	41–60	37	8.96	5.40
	60–80	10	12.10	10.90
Education level	Illiterate	20	6.75	4.42
	Elementary education	13	13.25	6.52
	Secondary education	10	12.90	6.15
	HSE	12	6.10	5.22
	Bachelor degree	8	6.10	5.22
	Higher education	7	11.70	6.41
Professions	THPs	14	23.55	12.4
	Laborers	03	4.95	5.10
	Teachers	05	7.01	7.90
	Midwives	06	14.2	9.83
	Housewives	15	6.65	6.10
	Herders	07	9.18	8.62
	Farmers	12	5.45	4.70
	Shopkeepers	03	4.55	3.33
	Students	05	5.02	4.18

No of Inf.: Number of informants, ANSRI: Average number of species reported by each informant, ANURI. Average number of uses reported by each informant, HSE: Higher Secondary Education, THPs: Traditional Health Practitioners.

**Table 2 plants-13-01379-t002:** Medicinal uses of the reported taxa in Tehsil Hajira (Pakistan) and their comparison with previous reports.

Scientific Name/Voucher Number	Local Name	Wild/Cultivated	Habit	Part Used	Method of Preparation	Mode of Application	Diseases Treated	Previous Use Reports
**Acanthaceae**								
*Dicliptera bupleuroides* Nees TJ-101	Churu	Wild	H	RT	AEX	External	Wounds	1•, 2•, 3•, 4∆, 5•, 6•, 7•, 8•, 9•, 10•, 11•, 12•, 13•, 14•, 15•, 16•, 17•, 18•, 19•, 20•, 21•, 22•
				LE	AEX	Internal	**Cough ***, Fever	
				LE	PD	External	Skin diseases *****, Eczema *	
*Justicia adhatoda* L. TJ-100	Baker	Wild	S	BA	PD	Internal	Stomachache	1■, 2•, 3•, 4•, 5•, 6•, 7•, 8•, 9•, 10•, 11•, 12•, 13∆, 14•, 15•, 16•, 17•, 18■, 19•, 20•, 21•, 22•
				LE	PD	Internal	**Constipation**	
				RT	AEX	Internal	Asthma, Cough	
*Pteracanthus urticifolius* (Wall. ex Kuntze) Bremek TJ-132	Herb	Wild	H	WP	DE	Internal	**Antiulcer**, Laxative, Diuretic, Rheumatism	1•, 2•, 3•, 4•, 5•, 6•, 7•, 8•, 9•, 10•, 11•, 12•, 13•, 14•, 15•, 16•, 17•, 18•, 19•, 20•, 21•, 22•
**Acoraceae**								
*Acorus calamus* L. TJ-13	Bach	Wild	H	RT	PD	Internal	**Digestive disorders**, Chronic dysentery	1•, 2•, 3•, 4•, 5■, 6•, 7•, 8∆, 9•, 10•, 11•, 12•, 13•, 14•, 15•, 16•, 17•, 18•, 19•, 20•, 21•, 22•
**Alliaceae**								
*Allium sativum* L. TJ-21	Lehsan	Cultivated	H	BL	RF	External	**Earache**	1•, 2•, 3•, 4•, 5•, 6•, 7•, 8•, 9∆, 10•, 11•, 12•, 13•, 14•, 15•, 16∆, 17•, 18∆, 19•, 20•, 21∆, 22•
					RF	Internal	Cough	
					AEX	Internal	Tuberculosis *	
**Amaranthaceae**								
*Achyranthes aspera* L. TJ-123	Puthkanda	Wild	H	WP	PA	External	Scorpion sting and snake bite	1∆, 2•, 3•, 4∆, 5■, 6•, 7•, 8∆, 9∆, 10•, 11•, 12•, 13•, 14∆, 15•, 16•, 17•, 18■, 19■, 20•, 21•, 22•
				ST	PD	Internal	**High fever**, Chest problem	
*Amaranthus viridis* L. TJ-114	Ganhar	Wild	H	LE	PA	External	Burning of feet	1•, 2•, 3∆, 4•, 5•, 6∆, 7•, 8∆, 9•, 10•, 11•, 12•, 13•, 14■, 15•, 16•, 17•, 18∆, 19■, 20•, 21∆, 22∆
				RT	PA	External	Scorpion sting and snake bite	
**Anacardiaceae**								
*Cotinus coggygria* Scop. TJ-98	Bahan	Wild	S	LE	DE	Internal	Hepatitis, Anemia, **Skin infections**	1•, 2•, 3•, 4•, 5•, 6■, 7•, 8•, 9•, 10•, 11•, 12•, 13•, 14•, 15•, 16•, 17•, 18•, 19•, 20•, 21•, 22•
				WP	AEX	Internal	Anti-ageing	
**Apocynaceae**								
*Carissa spinarum* L.TJ-48	Granada	Wild	S	FR, ST	PD	Internal	Pain, Inflammation	1■, 2•, 3•, 4•, 5•, 6∆, 7•, 8•, 9•, 10•, 11•, 12•, 13•, 14•, 15•, 16•, 17•, 18∆, 19•, 20•, 21•, 22•
				LE, SD	PD	Internal	Throat pain *	
				FR	ET	Internal	**Blood purifier**	
*Tylophora hirsuta* Wight TJ-113	Budhi bail	Wild	C	WP	PO	External	**Wound or burn relief**	1•, 2•, 3•, 4•, 5•, 6∆, 7•, 8•, 9•, 10•, 11•, 12•, 13•, 14•, 15•, 16•, 17•, 18•, 19•, 20•, 21•, 22•
				WP	DE	Internal	Asthma, High blood pressure, Diarrhea, Allergic conditions	
*Vinca major* Brot. TJ-41	Sada bhari booti	Wild	H	WP	IN	External	Nose bleeding, **Sore throat**, Mouth ulcers,	1•, 2•, 3•, 4•, 5•, 6•, 7•, 8•, 9•, 10•, 11•, 12•, 13•, 14•, 15•, 16•, 17•, 18•, 19•, 20•, 21•, 22•
				WP	AEX	internal	Hardening of arteries, Uterine bleeding	
**Araceae**								
*Arisaema utile* Hook. f. ex Schott TJ-134	Sanp ki khum	Wild	H	RH, LE	PA	External	**Snake bite**	1•, 2•, 3•, 4•, 5•, 6•, 7•, 8•, 9•, 10•, 11•, 12•, 13•, 14•, 15•, 16•, 17•, 18•, 19•, 20•, 21•, 22•
				RH	PA	External	Toothache	
**Araliaceae**								
*Hedera nepalensis* K. Koch TJ-05	Batkal	Wild	C	LE	PA	External	Burning sensation of hand and feet *	1■, 2■, 3•, 4•, 5■, 6∆, 7•, 8•, 9•, 10∆, 11•, 12•, 13•, 14•, 15∆, 16•, 17•, 18•, 19•, 20•, 21•, 22•
				LE	DE	Internal	**Diabetes**	
**Asparagaceae**								
*Asparagus gracilis* Salisb. TJ-11	Shatavari	Wild/cultivated	H	RT	AEX	Internal	Diarrhea, **Dysentery**	1•, 2•, 3•, 4•, 5•, 6•, 7•, 8•, 9•, 10•, 11•, 12•, 13•, 14•, 15•, 16•, 17•, 18•, 19•, 20•, 21•, 22•
				FR	PD	Internal	Aphrodisiac	
*Asparagus racemosus* Willd TJ-71	Sainsarbuti	Wild/cultivated	H	WP	AEX	Internal	Antidiarrheal, Antispasmodic, Diuretic	1•, 2•, 3•, 4•, 5•, 6•, 7•, 8•, 9•, 10•, 11•, 12•, 13•, 14•, 15•, 16•, 17•, 18•, 19•, 20•, 21∆, 22•
				RH	PD	Internal	Body weakness	
**Aspleniaceae**								
*Asplenium dalhousiae* Hook TJ-32	Naroky	Wild	H	WP	AEX	Internal	**Antibacterial**	1•, 2•, 3•, 4•, 5•, 6•, 7•, 8•, 9•, 10•, 11•, 12•, 13•, 14•, 15•, 16•, 17•, 18•, 19•, 20•, 21•, 22•
				WP	PA	External	Curing blisters	
				RH	DE	Internal	Gonorrhea, Hepatitis	
**Asteraceae**								
*Achillea millefolium* L. TJ-63	Kangi	Wild	H	RT, LE	JU	Internal	**Pneumonia ***	1■, 2■, 3•, 4∆, 5•, 6∆, 7∆, 8•, 9•, 10■, 11•, 12• 13•, 14•, 15•, 16•, 17•, 18•, 19•, 20•, 21•, 22•
				LE	PD	External	Toothache	
*Anaphalis margaritacea* L. TJ-31	Chitti buti	Wild	H	WP	PO	Internal	Diarrhea, **Pulmonary infections**	1•, 2•, 3•, 4•, 5•, 6•, 7■, 8•, 9•, 10•, 11•, 12•, 13•, 14• 15•, 16•, 17•, 18•, 19•, 20•, 21•, 22•
				WP	IN	Internal	Burns, Ulcer, Headache, Sores	
*Artemisia dubia* Wall. ex Bess. TJ-22	Asfanthene	Wild	H	LE	PA	External	**Cuts and wound**, Ear diseases	1■, 2■, 3•, 4•, 5•, 6•, 7•, 8•, 9•, 10•, 11•, 12•, 13•, 14•, 15•, 16•, 17•, 18•, 19•, 20•, 21•, 22∆
*Artemisia scoparia* Waldst. & Kit. TJ-129	Chahu	Wild	H	LE	IN	External	Body ache	1•, 2•, 3•, 4•, 5•, 6•, 7•, 8•, 9•, 10•, 11•, 12•, 13•, 14•, 15•, 16•, 17•, 18•, 19•, 20•, 21•, 22•
				WP	DE	Internal	Internal bleeding, Headache, **Cold**	
*Artemisia vulgaris* L. TJ-96	Chahu	Wild	H	LE	PD	Internal	**Oral thrush**, Tumors	1•, 2•, 3•, 4•, 5•, 6•, 7•, 8•, 9•, 10∆, 11•, 12•, 13•, 14•, 15•, 16•, 17•, 18•, 19•, 20•, 21•, 22•
*Bidens bipinnata* L. TJ-103	Bahanra buti	Wild	H	LE	AEX	External	Leprosy, Skin cut	1•, 2•, 3•, 4•, 5•, 6∆, 7•, 8•, 9•, 10•, 11•, 12•, 13•, 14•, 15•, 16•, 17•, 18•, 19•, 20•, 21•, 22•
*Bidens biternata* (Lour.) Merr. & Sherff TJ-95	Suryala	Wild	H	LE	IN	Internal	Sore throat	1•, 2•, 3•, 4•, 5•, 6■, 7•, 8•, 9•, 10•, 11•, 12•, 13•, 14•, 15•, 16•, 17•, 18•, 19•, 20•, 21•, 22•
				RT	PA	External	**Toothache**	
*Calendula officinalis* L. TJ-135	Sadberga	Cultivated	H	LE	JU	Internal	**Ear pain ***	1■, 2•, 3•, 4■, 5•, 6•, 7•, 8•, 9•, 10•, 11•, 12•, 13•, 14•, 15•, 16•, 17•, 18•, 19•, 20•, 21•, 22•
				YB	AEX	Internal	Kidney stone	
*Carthamus tinctorius* L. TJ-53	Kasumbaha	Cultivated	H	FL	JU	Internal	**Chicken pox**, **measles ***	1■, 2•, 3•, 4•, 5•, 6•, 7•, 8•, 9•, 10•, 11•, 12•, 13•, 14•, 15•, 16•, 17•, 18■, 19•, 20•, 21•, 22•
				FL	RB	External	Pneumonia	
*Cirsium vulgare* (Savi) Ten. TJ-84	Kandyara	Wild	H	RT	DE	Internal	Worms	1•, 2•, 3•, 4•, 5•, 6•, 7•, 8•, 9•, 10•, 11•, 12•, 13•, 14•, 15•, 16•, 17•, 18•, 19•, 20•, 21•, 22•
				RT	IN	Internal	**Tonic**, Diuretic, Astringent	
*Erigeron canadensis* L. TJ-105	Neeli buti	Wild	H	VP	DE	Internal	Diuretic, Astringent, **Diarrhea**, Dysentery	1•, 2•, 3•, 4•, 5•, 6•, 7•, 8•, 9■, 10•, 11•, 12•, 13•, 14•, 15•, 16•, 17•, 18•, 19•, 20•, 21•, 22•
*Gerbera gossypina* (Royle) Beauverd TJ-136	Put potula	Wild	H	LE	PA	External	Skin diseases, Cuts, Bone fracture	1■, 2∆, 3•, 4•, 5•, 6•, 7•, 8•, 9•, 10•, 11•, 12•, 13•, 14•, 15•, 16•, 17•, 18•, 19•, 20•, 21•, 22•
				AP	TE	Internal	**Nerve disorders**	
*Inula cappa* (Buch.-Ham.ex D. Don) DC TJ-104.	Guliston	Wild	H	RT	JU	Internal	Peptic ulcers, Indigestion, **Gastric disorders**	1•, 2•, 3•, 4•, 5•, 6•, 7•, 8•, 9•, 10•, 11•, 12•, 13•, 14•, 15•, 16•, 17•, 18•, 19•, 20•, 21•, 22•
				RT	DE	Internal	Fever, Miscarriage	
*Parthenium hysterophorus* L. TJ-12	Gandi booti	Wild	H	RT	DE	Internal	Eczema	1■, 2■, 3•, 4•, 5•, 6•, 7•, 8•, 9•, 10•, 11•, 12•, 13•, 14•, 15•, 16•, 17•, 18•, 19∆, 20•, 21•, 22•
*Tagetes minuta* L. SSTJ-33	Sadberga	Wild/cultivated	H	LE	JU	Internal	**Earache**	1•, 2•, 3•, 4•, 5•, 6•, 7•, 8•, 9•, 10•, 11•, 12•, 13•, 14•, 15•, 16•, 17•, 18∆, 19•, 20•, 21•, 22•
*Taraxacum officinale* Weber ex F.H. Wigg. TJ-125	Hand	Wild/cultivated	H	RT	PA	External	Scorpion sting *	1∆, 2∆, 3•, 4•, 5•, 6■, 7∆, 8•, 9•, 10∆, 11•, 12•, 13•, 14•, 15•, 16•, 17•, 18∆, 19•, 20•, 21∆, 22•
				WP	DE	Internal	**Jaundice**, Constipation, Chronic disorders of kidney and liver	
**Balsaminaceae**								
*Impatiens bicolor* Royle TJ-90	Tilcawli	Wild	H	SD	PD	Internal	**Tonic**, Diuretic	1•, 2•, 3•, 4•, 5•, 6•, 7•, 8•, 9•, 10•, 11•, 12•, 13•, 14•, 15•, 16•, 17•, 18•, 19•, 20•, 21•, 22•
				LE	PA	External	Joints pain	
**Berberidaceae**								
*Berberis lycium* Royle TJ-19	Sumblu	Wild	S	RT	AEX	Internal	Diabetes, Jaundice	1■, 2■, 3•, 4∆, 5∆, 6∆, 7•, 8∆, 9∆, 10∆, 11•, 12•, 13•, 14•, 15∆, 16•, 17∆, 18■, 19•, 20•, 21∆, 22∆
				RT	IN	Internal	Chronic diarrhea	
				RT	PA	External	Skin diseases	
**Brassicaceae**								
*Brassica compestris* L. TJ-43	Sarsuu	Cultivated	H	SD	PA	External	Skin infection *	1•, 2•, 3•, 4•, 5•, 6•, 7•, 8•, 9∆, 10•, 11•, 12•, 13•, 14•, 15•, 16•, 17•, 18•, 19•, 20•, 21•, 22•
				SD, LE	PO	External	Backache *, **Joints pain ***	
*Capsella bursa-pastoris* (L.) Medic. TJ-83	Kangani	Wild	H	AP	CK	Internal	Diarrhea *	1•, 2•, 3•, 4•, 5•, 6■, 7•, 8•, 9•, 10•, 11∆, 12•, 13•, 14•, 15•, 16•, 17•, 18•, 19•, 20•, 21•, 22•
				RT	PA	External	**Psoriasis ***	
				SD	PD	Internal	Cough, cold and fever	
**Buxaceae**								
*Sarcococca saligna* Müll. Arg. TJ-139	Nadroon	Wild	S	LE, SH	DE	Internal	Joints pain, **Blood purifier**, **Purgative**	1■, 2■, 3•, 4•, 5•, 6•, 7•, 8•, 9•, 10∆, 11•, 12•, 13•, 14•, 15•, 16•, 17•, 18•, 19•, 20•, 21•, 22•
				LE	PD	External	Burns	
**Cannabaceae**								
*Cannabis sativa* L. TJ-35	Bhang	Wild	H	LE	PA	External	**Swelling joints**	1•, 2•, 3∆, 4∆, 5•, 6•, 7•, 8∆, 9∆, 10•, 11•, 12•, 13•, 14∆, 15∆, 16•, 17∆, 18■, 19■, 20•, 21∆, 22∆
				LE	PA	External	Leeches, Lice	
**Chenopodiaceae**								
*Chenopodium album* L. TJ-128	Bathu	Wild	H	LE	CK	Internal	Constipation, Intestinal worms *	1•, 2•, 3•, 4•, 5•, 6∆, 7∆, 8•, 9•, 10∆, 11•, 12•, 13•, 14∆, 15•, 16•, 17∆, 18■, 19∆, 20•, 21∆, 22∆
				LE	JU	Internal	Jaundice, Urinary disorders	
				WP	PO	External	**Swollen feet**	
*Dysphania ambrosioides* (L.) Mosyakin & Clemants TJ-62	Challa Baathu	Wild	H	LE	BO	Internal	Piles, Gas trouble *, Stomach griping, **Indigestion**	1•, 2•, 3•, 4•, 5•, 6•, 7•, 8•, 9•, 10•, 11•, 12•, 13•, 14∆, 15•, 16•, 17•, 18•, 19■, 20•, 21∆, 22•
**Colchicaceae**								
*Colchicum luteum* Baker TJ-42	Suranjan	Wild	H	CO	PD	Internal	Spleen, **Liver disease**, Blood purifier, Diuretic	1•, 2•, 3•, 4∆, 5•, 6•, 7•, 8•, 9•, 10•, 11•, 12•, 13•, 14•, 15•, 16•, 17•, 18•, 19•, 20•, 21•, 22•
				RT	PA	External	Gout, Arthritic pain	
**Convolvulaceae**								
*Convolvulus arvensis* L. TJ-72	Hiranpadi	Wild	C	RT	PD	Internal	Laxative	1•, 2•, 3•, 4•, 5•, 6•, 7•, 8•, 9•, 10•, 11•, 12•, 13•, 14∆, 15•, 16•, 17∆, 18•, 19∆, 20•, 21•, 22∆
				WP	AEX	Internal	**Diabetes ***	
				WP	JU	Internal	Constipation	
*Ipomoea purpurea* (L.) Roth TJ-118	Earh	Cultivated	C	SD	PD	Internal	Purgative, **Tonic**	1•, 2•, 3•, 4•, 5•, 6•, 7•, 8•, 9•, 10•, 11•, 12•, 13•, 14•, 15•, 16•, 17•, 18•, 19•, 20•, 21•, 22•
				WP	AEX	External	**Skin disorders**	
**Cucurbitaceae**								
*Momordica charanta* L. TJ-04	Karella	Cultivated	C	FR	JU	Internal	**Diabetes**, Piles	1■, 2•, 3∆, 4•, 5•, 6•, 7•, 8•, 9•, 10•, 11•, 12•, 13•, 14•, 15•, 16∆, 17•, 18■, 19•, 20•, 21∆, 22•
				LE	PA	External	Inflammation	
*Solena heterophylla* Lour. TJ-122	Bankakri	Wild	C	LE	PA	external	**Inflamed skin**	1•, 2•, 3•, 4•, 5•, 6•, 7•, 8•, 9•, 10•, 11•, 12•, 13•, 14•, 15•, 16•, 17•, 18•, 19•, 20•, 21•, 22•
				RT	JU	Internal	Dysuria, Spermatorrhea	
**Cuscutaceae**								
*Cuscuta reflexa* Roxb. TJ-144	Neela dahri	Wild	C	WP	JU	Internal	Jaundice	1■, 2•, 3•, 4•, 5•, 6•, 7•, 8•, 9•, 10•, 11•, 12•, 13•, 14•, 15•, 16•, 17•, 18■, 19∆, 20•, 21•, 22•
				WP	AEX	External	**Dandruff**	
**Cyperaceae**								
*Cyperus rotundus* L. TJ-82	Mutharr	Wild	H	RH	PA	External	Snake bite *	1•, 2•, 3•, 4•, 5•, 6■, 7•, 8•, 9•, 10•, 11•, 12•, 13•, 14∆, 15•, 16•, 17•, 18•, 19∆, 20•, 21•, 22•
				WP	AEX	Internal	Nausea, **Inflammation**, Fever	
**Dioscoraceae**								
*Dioscorea deltoidea* Wall ex Kunth TJ-79	Saki ganda	Wild	C	TU	PD	Internal	Excretion, Constipation, Tonic, Productive cough, Worms’ expulsion	1∆, 2∆, 3•, 4∆, 5•, 6•, 7•, 8•, 9•, 10•, 11•, 12•, 13•, 14•, 15•, 16•, 17•, 18•, 19•, 20•, 21•, 22•
				TU	PA	Internal	**Intestinal worms**	
**Dryopteridaceae**								
*Dryopteris filix-mas* (L.) Schott TJ-66	Kungi	Wild	H	RT	PD	Internal	Purgative	1•, 2•, 3•, 4•, 5•, 6•, 7•, 8•, 9•, 10•, 11•, 12•, 13•, 14•, 15•, 16•, 17•, 18•, 19•, 20•, 21•, 22•
				RT	AEX	Internal	**Tapeworm ***	
**Ebenaceae**								
*Diospyros lotus* L. TJ-59	Amluk	Wild	T	FR	ET	Internal	**Constipation ***	1•, 2•, 3•, 4■, 5•, 6•, 7•, 8•, 9•, 10•, 11•, 12•, 13•, 14•, 15•, 16•, 17•, 18•, 19•, 20•, 21∆, 22•
				FR	ET	Internal	Purgative, Laxative	
**Elaeagnaceae**								
*Elaeagnus umbellata* Thunb TJ-37	Kankoli	Wild	S	LE, FL	DE	Internal	Heart diseases, Cough	1■, 2■, 3•, 4•, 5•, 6•, 7•, 8•, 9•, 10•, 11•, 12•, 13•, 14•, 15•, 16•, 17•, 18•, 19•, 20•, 21•, 22•
				SD	RF	Internal	Immunity	
				BR	PD	External	**Toothache**	
**Euphorbiaceae**								
*Euphorbia helioscopia* L. TJ-56	Doodle	Wild	H	ST	LX	External	**Ring worms ***, Laxative	1∆, 2∆, 3∆, 4•, 5•, 6∆, 7•, 8•, 9•, 10•, 11•, 12•, 13•, 14■, 15•, 16•, 17•, 18∆, 19■, 20•, 21•, 22∆
				SD	JU	Internal	Cholera	
				SD	PD	internal	Constipation	
*Mallotus philippensis* (Lam.) Müll.Arg. TJ-100	Kamilla	Wild	T	FR	PD	External	Tapeworm *, Skin diseases, Mumps measles	1•, 2•, 3•, 4∆, 5•, 6∆, 7•, 8•, 9•, 10•, 11•, 12•, 13•, 14•, 15•, 16•, 17•, 18∆, 19•, 20•, 21•, 22•
*Ricinus communis* L. TJ-119	Harnoli	Wild	S	LE	PA	External	**Joints pain**	1•, 2•, 3∆, 4∆, 5■, 6•, 7•, 8•, 9•, 10∆, 11•, 12•, 13•, 14•, 15•, 16•, 17•, 18∆, 19•, 20•, 21•, 22∆
				LE	DE	Internal	Jaundice, Constipation	
**Fabaceae**								
*Vachellia nilotica* (L.) P.J.H. Hurter and Mabb. TJ-91	Kiker	Wild/cultivated	T	BE/RT	PD	Internal	Back pain	1•, 2•, 3•, 4•, 5•, 6•, 7•, 8•, 9•, 10•, 11•, 12∆, 13•, 14■, 15•, 16•, 17•, 18•, 19■, 20•, 21•, 22•
				BA	DE	Internal	**Stomach disorder**	
*Astragalus psilocentros* Fisch TJ-36	Tindni	Wild	S	RT	PD	Internal	Hepatitis, Heart diseases, Regulate immune system	**1•, 2•, 3•, 4•, 5•, 6•, 7•, 8•, 9•, 10•, 11•, 12•, 13•, 14•, 15•, 16•, 17•, 18•, 19•, 20•, 21•, 22•**
				RT	PA	External	**Healing wounds**	
				WP	AEX	Internal	Diabetes, Chronic asthma, Diarrhea, Vomiting, Bone marrow	
*Bauhinia variegata* L. TJ-47	Kachnar	Cultivated	S	LE, FL	PA	Internal	Diarrhea	1•, 2•, 3•, 4•, 5•, 6•, 7•, 8•, 9•, 10•, 11•, 12•, 13•, 14•, 15•, 16•, 17•, 18•, 19•, 20•, 21•, 22•
				BA	DE		**Skin diseases**	
*Desmodium elegans* DC. TJ-75	Thalbai	Wild	S	RT	PD	External	Scorpion sting and snake bite	1•, 2•, 3•, 4•, 5•, 6•, 7•, 8•, 9•, 10•, 11•, 12•, 13•, 14•, 15•, 16•, 17•, 18•, 19•, 20•, 21•, 22•
				RT	AEX	Internal	Carminative, Diuretic, Tonic	
				BA	JU	Internal	Peptic ulcer	
*Indigofera heterantha* Brandis TJ-87	Kanthi	Wild	S	WP	PD	Internal	Hepatitis, Whooping’s *, Cough *	1•, 2•, 3•, 4•, 5•, 6∆, 7■, 8•, 9•, 10•, 11•, 12•, 13•, 14•, 15∆, 16•, 17•, 18•, 19•, 20•, 21•, 22•
				RT	AEX	External	Dying of blackening of hair *	
*Lespedeza juncea* (L. f.) Pers. TJ-99	Kuchani	Wild	S	RT	PA	External	**Snake bite**	1•, 2•, 3•, 4•, 5•, 6•, 7•, 8•, 9•, 10•, 11•, 12•, 13•, 14•, 15•, 16•, 17•, 18•, 19•, 20•, 21•, 22•
				ST	DE	Internal	Neuralgia, Rheumatism	
*Mucuna pruriens* (L.) DC. TJ-52	Kunch kuri	Wild	H	BE	PA	External	**Scorpion sting**	1•, 2•, 3•, 4•, 5•, 6•, 7•, 8•, 9•, 10•, 11•, 12•, 13•, 14•, 15•, 16•, 17•, 18•, 19•, 20•, 21•, 22•
				SD	AEX	Internal	Parkinson’s disease	
				WP	AEX	Internal	Neuroprotective, Analgesic	
*Robinia pseudoacacia* L. TJ-137	Kikar	Wild	T	BA	CH	External	**Toothache**	1■, 2■, 3•, 4•, 5•, 6•, 7•, 8•, 9•, 10•, 11•, 12•, 13•, 14•, 15•, 16•, 17•, 18•, 19•, 20•, 21•, 22•
**Gentianaceae**								
*Swertia angustifolia* Buch.-Ham. ex D. Don TJ-117	Cheratbotay	Wild	H	WP	AEX	internal	**Malarial fever**, Bronchial asthma, Blood purification, Febrifuge	1•, 2•, 3•, 4•, 5•, 6•, 7•, 8•, 9•, 10•, 11•, 12•, 13•, 14•, 15•, 16•, 17•, 18•, 19•, 20•, 21•, 22•
*Swertia cordata* (Wall. ex G. Don) C.B. Clarke TJ-65	Charita	Wild	H	WP	DE, JU	Internal	Fever, Jaundice, Indigestion, Cough, Cold, Typhoid	1•, 2•, 3•, 4•, 5•, 6•, 7■, 8•, 9•, 10•, 11•, 12•, 13•, 14•, 15•, 16•, 17•, 18•, 19•, 20•, 21•, 22•
**Geraniaceae**								
*Geranium wallichianum* D. Don.ex Sweet TJ-133	Ratanjot	Wild	H	RH	PD,	Internal	Backache *	1•, 2•, 3•, 4∆, 5∆, 6•, 7■, 8•, 9•, 10∆, 11•, 12•, 13•, 14•, 15•, 16•, 17∆, 18•, 19•, 20•, 21•, 22•
					CK	Internal	Mouth ulcer *, Chronic diarrhea, High blood pressure, Stomach disorders *	
				LE	PA	External	Joints pain, **Toothache**	
				RT	JU	Internal	Jaundice, Diarrhea	
**Hypericaceae**								
*Hypericum perforatum* L. TJ-50	Chamba	Wild	H	LE, FL	DE	Internal	Carminative *, Stimulant *	1∆, 2∆, 3•, 4•, 5•, 6•, 7•, 8∆, 9•, 10•, 11∆, 12•, 13•, 14•, 15•, 16•, 17•, 18•, 19•, 20•, 21•, 22•
				LE	DE	Internal	**Diuretic**	
				WP	AEX	External	Wounds, Bruises *	
**Juglandaceae**								
*Juglans regia* L. TJ-9	Akhroat	Cultivated/wild	T	FR	ET	Internal	Brain weakness *	1■, 2■, 3∆, 4∆, 5•, 6•, 7•, 8∆, 9•, 10∆, 11∆, 12•, 13•, 14•, 15∆, 16•, 17∆, 18∆, 19•, 20•, 21∆, 22•
				RT, LE	PD	External	Antiseptic	
				SD	Oil	External	Rheumatism	
**Lamiaceae**								
*Ajuga bracteosa* Wall. ex Benth. TJ-102	Rati buti	Wild	H	LE	AEX	Internal	Earache, Throat pain	1■, 2■, 3•, 4•, 5∆, 6∆, 7•, 8∆, 9■, 10∆, 11•, 12•, 13•, 14•, 15•, 16•, 17•, 18•, 19•, 20•, 21•, 22■
				AP	AEX	Internal	**Blood purification**, Pimples	
*Caryopteris odorata* (D. Don) B.L .Rob. TJ-01	Bahata jari		H	LE	PD	External	Foot ulcer*, Wounds	1•, 2•, 3•, 4∆, 5•, 6•, 7•, 8•, 9•, 10•, 11•, 12•, 13•, 14•, 15•, 16•, 17•, 18•, 19•, 20•, 21•, 22•
				FL	PD	internal	Diabetic*, Tumors*	
				LE, BA	PD	Internal	Nausea*, Vomiting*, Abdominal pain*	
*Mentha longifolia* (L.) L. TJ-16	Pudenda	Cultivated/wild	H	LE	PD	Internal	**Digestive disorders**	1•, 2•, 3■, 4•, 5■, 6•, 7•, 8■, 9∆, 10∆, 11•, 12•, 13•, 14•, 15•, 16■, 17∆, 18∆, 19•, 20■, 21∆, 22∆
				LE, SH, FP	PD	Internal	Joints pain	
*Micromeria biflora* (Buch.-Ham. ex D. Don) Benth. TJ-69	Boine	Wild	H	ST	JU	Internal	Urinary disorders	1•, 2•, 3•, 4•, 5∆, 6■, 7•, 8•, 9•, 10•, 11•, 12•, 13•, 14•, 15•, 16•, 17•, 18•, 19•, 20•, 21•, 22•
				LE	DE	Internal	**Digestive disorders**	
				WP	AEX	Internal	Diuretic, Vomiting, Constipation, Headache	
*Nepeta laevigata* D. Don Hand.-Mazz. TJ-78	Badrn boya	Wild	H	FR	IN	Internal	**Dysentery**	1■, 2■, 3•, 4•, 5•, 6•, 7•, 8•, 9•, 10•, 11•, 12•, 13•, 14•, 15•, 16•, 17•, 18•, 19•, 20•, 21•, 22•
*Origanum vulgare* L. TJ-92	Sahthar	Wild	H	LE	CH	External	Toothache, **Mouth gum***	1•, 2•, 3•, 4•, 5■, 6■, 7∆, 8∆, 9•, 10•, 11•, 12•, 13•, 14•, 15∆, 16•, 17•, 18•, 19•, 20•, 21•, 22•
				WP	DE	Internal	Skin infection, Digestive disorders	
*Otostegia limbata* (Benth.)Boiss. TJ-97	Chiti ptaki	Wild	S	LE	PD	Internal	**Gum diseases**	1•, 2•, 3•, 4•, 5■, 6■, 7•, 8■, 9∆, 10•, 11•, 12•, 13•, 14•, 15•, 16•, 17•, 18∆, 19•, 20•, 21•, 22•
				LE	AEX	Internal	Wounds, Mouth ulcer, Skin and eye diseases	
*Vitex negundo* L. TJ-115	Bana	Wild	S	LE	AEX	Internal	Urinary disorders, Mild fever	1•, 2•, 3•, 4∆, 5■, 6•, 7•, 8•, 9•, 10•, 11•, 12•, 13•, 14•, 15•, 16•, 17•, 18∆, 19•, 20•, 21•, 22•
**Lycopodiaceae**								
*Lycopodium japonicum* Thunb. TJ-74	Bhanjabasa	Wild	H	WP	PD	External	Wounds healing	1•, 2•, 3•, 4•, 5•, 6•, 7•, 8•, 9•, 10•, 11•, 12•, 13•, 14•, 15•, 16•, 17•, 18•, 19•, 20•, 21•, 22•
				RT	AEX	External	**Body aches**, Swelling	
**Lythraceae**								
*Punica granatum* L. TJ-18	Anardana	Cultivated	S	FR	JU	Internal	**Jaundice**, Heam synthesis *	1■, 2■, 3∆, 4∆, 5•, 6∆, 7•, 8∆, 9•, 10•, 11∆, 12•, 13•, 14•, 15•, 16∆, 17∆, 18∆, 19•, 20•, 21∆. 22•
				LE	JU	Internal	Dysentery	
				FR	RF	Internal	Cough, Tonic	
				BA, ST, RT	DE	Internal	Mouthwash	
				RT	DE	Internal	Expectorant	
**Malvaceae**								
*Grewia asiatica* L. TJ-130	Phalsa	Cultivated	T	RT, BA	IN	Internal	Febrifuge, Diarrhea	1•, 2•, 3•, 4•, 5•, 6•, 7•, 8•, 9•, 10•, 11•, 12•, 13•, 14•, 15•, 16•, 17•, 18•, 19•, 20•, 21•, 22•
				LE	DE	Internal	Urinary tract infection, Sexually transmitted diseases	
				BA, RT	PA	External	**Rheumatism**, Arthritis	
				FR	PD	Internal	Astringent, Stomachache, Burning sensation, Fever	
*Malvastrum coromandelianum* (L.) Garcke TJ-76	Bariar	Wild	H	AP	DE	Internal	Kill worms, Dysentery	1■, 2■, 3•, 4•, 5•, 6■, 7•, 8•, 9•, 10•, 11•, 12•, 13•, 14•, 15•, 16•, 17•, 18•, 19∆, 20•, 21•, 22•
				FL	DE	Internal	Fever	
				LE	PA	External	**Wound healing**	
**Martyniaceae**								
*Martynia annua* L. TJ-131	Jawahta jori	Wild	H	SD	Oil	External	Itching, Skin infection	1•, 2•, 3•, 4•, 5•, 6•, 7•, 8•, 9•, 10•, 11•, 12•, 13•, 14•, 15•, 16•, 17•, 18•, 19•, 20•, 21•, 22•
				LE	PA	External	**Wound healing**	
				WP	AEX	Internal	Epilepsy, Inflammation, Tuberculosis, Sore throat	
**Meliaceae**								
*Cedrela toona* Roxb. ex Rottler and Wild TJ-108	Toon	Wild	T	LE	PD	Internal	Fever, Diabetes, **Skin diseases**, Blood purifier	1•, 2•, 3•, 4•, 5•, 6•, 7•, 8•, 9•, 10•, 11•, 12•, 13•, 14•, 15•, 16•, 17•, 18•, 19•, 20•, 21•, 22•
				BA	PD	Internal	Dysentery	
				BA	PD	External	Healing wounds	
*Melia azedarach* L. TJ-29	Dreak	Wild/cultivated	T	LE	JU	Internal	**Malaria**, Typhoid *	1•, 2•, 3∆, 4∆, 5∆, 6•, 7•, 8•, 9•, 10•, 11•, 12•, 13∆, 14■, 15•, 16•, 17•, 18∆, 19■, 20•, 21∆, 22•
				LE	PD	Internal	Urinary disorders	
				LE, FR	PA	External	Wounds	
**Menispermaceae**								
*Cissampelos pareira* L. TJ-02	Batrarr	Wild	C	LE	PO	External	**Snake bite**, Dropsy	1•, 2•, 3•, 4•, 5•, 6■, 7•, 8•, 9•, 10•, 11•, 12•, 13•, 14•, 15•, 16•, 17•, 18•, 19•, 20•, 21•, 22•
				WP	AEX	Internal	Diarrhea, Stomach diseases	
**Moraceae**								
*Ficus carica* L. TJ-28	Anjeer	Wild/cultivated	T	FR	ET	Internal	**Constipation**, Urinary bladder problem, Piles, Anemia	1■, 2■, 3∆, 4•, 5•, 6•, 7•, 8∆, 9•, 10•, 11∆, 12•, 13•, 14•, 15•, 16∆, 17∆, 18•, 19•, 20•, 21∆, 22∆
				LE	LX	External	Nail wounds	
				LX	RB	External	Extract spines from feet or other body organs	
*Ficus palmata* Forssk. TJ-30	Phagwar	Wild	T	FR	RF	Internal	**Diseases of lungs and bladder**	1■, 2■, 3•, 4•, 5•, 6•, 7•, 8•, 9•, 10•, 11•, 12•, 13•, 14•, 15•, 16•, 17•, 18∆, 19•, 20•, 21•, 22•
				AP	PA	External	Freckles	
*Morus alba* L. TJ-109	Toot	Cultivated	T	FR	JU	Internal	Throat pain *, **Tonsils**	1•, 2•, 3•, 4•, 5•, 6•, 7•, 8•, 9•, 10•, 11•, 12•, 13•, 14■, 15•, 16•, 17∆, 18•, 19■, 20•, 21∆, 22•
				FR	ET	Internal	Cough, Chest problem *	
**Nyctaginaceae**								
*Mirabilis jalapa* L. TJ-110	Altaa	Wild/cultivated	H	LE	CK	Internal	Jaundice, **Dropsy**	1•, 2•, 3∆, 4•, 5•, 6•, 7•, 8•, 9•, 10•, 11•, 12•, 13•, 14•, 15•, 16•, 17•, 18•, 19•, 20•, 21•, 22∆
				FL	PD	Internal	Piles	
				LE	PA	External	Wounds	
**Oleaceae**								
*Jasminum officinale* L. TJ-86	Chambeli	Cultivated	S	LE	PA	External	Scorpion sting and snake bite	1•, 2•, 3•, 4•, 5•, 6∆, 7•, 8•, 9•, 10•, 11•, 12•, 13•, 14∆, 15•, 16•, 17•, 18•, 19∆, 20•, 21•, 22•
				LE	DE	External	**Toothache**	
*Ligustrum lucidum* W.T. Aiton TJ-138	Guliston	Wild	S	AP	AEX	Internal	**Antitumor**	1∆, 2∆, 3•, 4•, 5•, 6•, 7•, 8•, 9•, 10•, 11•, 12•, 13•, 14•, 15•, 16•, 17•, 18•, 19•, 20•, 21•, 22•
				FR	PD	Internal	Hypertension *, Hepatitis *, Diuretic *, Tonic *, Antibacterial *, Antiseptic *	
				LE	DE	Internal	Febrifuge *	
				BA, ST	IN	Internal	Diaphoretic *	
*Olea ferruginea* Wall. ex Aitch. TJ-10	Kove	Cultivated	T	LE	PD	Internal	**Mouth ulcer**, Skin diseases	1•, 2•, 3•, 4∆, 5∆, 6■, 7•, 8∆, 9•, 10∆, 11•, 12•, 13•, 14•, 15•, 16•, 17•, 18∆, 19•, 20•, 21∆, 22•
				FR	Oil	External	Hair growth *	
				LE, BA	TE	Internal	Cold, Cough, Flu	
**Orchidaceae**								
*Habenaria digitata* Lindl. TJ-111	Hirvi	Wild	H	LE	PA	External	Snake bite	1•, 2•, 3•, 4•, 5•, 6•, 7•, 8•, 9•, 10•, 11•, 12•, 13•, 14•, 15•, 16•, 17•, 18•, 19•, 20•, 21•, 22•
				TU	JU	Internal	Fever, **Asthma**, Skin diseases, Cough	
**Oxalidaceae**								
*Oxalis corniculata* L. TJ-85	Khati buti	Wild	H	WP	PA	External	**Eye pain**, Vitiligo *****	1•, 2•, 3∆, 4•, 5∆, 6∆, 7∆, 8•, 9•, 10•, 11•, 12•, 13•, 14∆, 15∆, 16•, 17•, 18∆, 19∆, 20•, 21∆, 22∆
				LE	JU	Internal	Jaundice, Dysentery, Fever	
				LE	PA	External	Worm *, Scorpion sting *	
**Pinaceae**								
*Pinus roxburghii* Sarg. TJ-88	Cheer	Cultivated	T	WO, RS	PD	External	Scorpion sting and snake bite	1■, 2■, 3•, 4•, 5•, 6∆, 7•, 8∆, 9•, 10•, 11•, 12•, 13•, 14•, 15•, 16•, 17•, 18∆, 19•, 20•, 21•, 22•
				LE, BA	PD	Internal	Dysentery	
*Pinus wallichiana* A.B. Jacks. TJ-80	Rahar	Cultivated/wild	T	RS	PO	External	**Wounds**	1∆, 2∆, 3•, 4•, 5•, 6•, 7■, 8•, 9∆, 10•, 11•, 12•, 13•, 14•, 15∆, 16•, 17•, 18•, 19•, 20•, 21∆, 22•
				BA, LE, SD	IN	internal	Cough, Fever, Asthma	
**Piperaceae**								
*Piper nigrum* L. TJ-27	Kalimirch	Cultivated	S	SD	PD	Internal	**Cough**, Asthma	1•, 2•, 3•, 4•, 5•, 6•, 7•, 8•, 9•, 10•, 11•, 12•, 13•, 14•, 15•, 16•, 17•, 18•, 19•, 20•, 21•, 22•
				SD	DE	Internal	Diarrhea, Flu	
				SD	PD	External	Toothache	
**Plantaginaceae**								
*Plantago lanceolata* L. TJ-45	Ispagol	Wild	H	LE	PA	External	Wounds	1■, 2■, 3∆, 4•, 5■, 6■, 7∆, 8∆, 9•, 10•, 11•, 12•, 13•, 14•, 15• 16•, 17∆, 18•, 19•, 20•, 21∆, 22•
				SD	AEX	Internal	**Dysentery**	
						External	Toothache	
*Plantago major* L. TJ-54	Achar	Wild	H	WP	PD	External	Infected hooves *	1•, 2•, 3•, 4•, 5•, 6•, 7■, 8•, 9•, 10∆, 11∆, 12•, 13•, 14•, 15•, 16•, 17•, 18∆, 19•, 20∆, 21∆, 22•
				SD, LE	AEX	Internal	Diarrhea, Asthma, Ulcer, Skin inflammation *	
**Poaceae**								
*Cynodon dactylon* (L.) Pers. TJ-142	Khabal	Wild	H	WP	PA	External	Knee sprain *, Wound	1•, 2•, 3•, 4•, 5•, 6■, 7•, 8•, 9•, 10•, 11•, 12•, 13•, 14∆, 15•, 16•, 17•, 18∆, 19∆, 20•, 21• 22∆
				WP	DE	Internal	Diuretic	
*Eleusine indica* (L.) Gaertn. TJ-44	Madhana gass	Wild	H	WP	DE	Internal	Dysuria, **Fever**, Jaundice, Rheumatism, infantile, Indigestion	1•, 2•, 3•, 4•, 5•, 6•, 7•, 8•, 9•, 10•, 11•, 12•, 13•, 14•, 15•, 16∆, 17•, 18•, 19∆, 20•, 21•, 22•
				WP	PA	External	Centipede and scorpion poisons	
				LE	AEX	Internal	Diuretic	
*Oplismenus undulatifolius* (Ard.) P. Beauv. TJ-140	Tukri ghass	Wild	H	LE	PD	Internal	Aphrodisiac	1•, 2•, 3•, 4•, 5•, 6•, 7•, 8•, 9•, 10•, 11•, 12•, 13•, 14•, 15•, 16•, 17•, 18•, 19•, 20•, 21•, 22•
				LE	PD	External	Sore, Snake bite	
				LE	AEX	Internal	**Painkiller**	
**Polygonaceae**								
*Polygonum aviculare* L. TJ-93	Bandky	Wild	H	WP	PO	External	Swollen, Inflamed areas	1•, 2•, 3•, 4•, 5•, 6•, 7•, 8•, 9•, 10•, 11•, 12•, 13•, 14•, 15•, 16•, 17•, 18•, 19•, 20•, 21•, 22•
				WP	DE	Internal	Diarrhea, Dyspepsia, **Itching skin**,	
				LE	CK	Internal	Anti-inflammatory, Astringent, Diuretic	
*Rumex dentatus* L. TJ-143	Alfari	Wild	H	RT	PA	External	Skin problem	1■, 2■, 3•, 4•, 5•, 6•, 7•, 8•, 9•, 10•, 11•, 12•, 13•, 14■, 15•, 16•, 17•, 18•, 19■, 20•, 21∆, 22■
				LE	PA	External	**Wounds**	
*Rumex hastatus* D. Don TJ-24	Khatimal	Wild	H	LE, RT	AEX	Internal	**Jaundice**	1∆, 2∆, 3•, 4∆, 5∆, 6•, 7•, 8•, 9•, 10•, 11•, 12•, 13•, 14•, 15•, 16•, 17•. 18∆, 19•, 20•, 21•, 22•
				LE, SH	DE	Internal	Urinary disorders, Constipation	
**Primulaceae**								
*Myrsine africana* L. TJ-57	Kathi	Wild	H	LE	DE	Internal	Blood purifier	1■, 2■, 3•, 4•, 5•, 6■, 7•, 8∆, 9•, 10•, 11•, 12•, 13•, 14•, 15•, 16•, 17•, 18•, 19•, 20•, 21•, 22∆
				FR	PD	Internal	**Stomach tonic**, Laxative	
**Pteridaceae**								
*Adiantum capillus-veneris* L. TJ-116	Ratanjot	Wild	H	FD	PA	External	Scorpion sting and snake bite *	1■, 2■, 3•, 4•, 5•, 6•, 7•, 8•, 9•, 10•, 11•, 12•, 13•, 14•, 15∆, 16•, 17•, 18•, 19•, 20•, 21∆, 22•
				LE	DE	Internal	**Cough**, Jaundice, Chest pain, Asthma	
*Adiantum incisum* Forssk. TJ-03	Barheipani	Wild	H	LE	JU	Internal	Cough, Body ache, Scabies	1■, 2∆, 3•, 4•, 5•, 6■, 7•, 8•, 9•, 10•, 11•, 12•, 13•, 14•, 15•, 16•, 17•, 18•, 19•, 20•, 21•, 22•
				LE	IN	Internal	Bronchitis, Body Weakness	
*Dryopteris filix-mas* (L.) Schott TJ-66	Kungi	Wild	H	RT	PD	Internal	Purgative	1•, 2•, 3•, 4•, 5•, 6•, 7•, 8•, 9•, 10•, 11•, 12•, 13•, 14•, 15•, 16•, 17•, 18•, 19•, 20•, 21•, 22•
				RT	AEX	Internal	**Tapeworm ***	
*Onychium japonicum* (Thunb.) Kunze TJ-94	Pathba	Wild	H	LE	JU	Internal	Dysentery	1•, 2•, 3•, 4•, 5•, 6•, 7•, 8•, 9•, 10•, 11•, 12•, 13•, 14•, 15•, 16•, 17•, 18•, 19•, 20•, 21•, 22•
				RH	JU	Internal	**Diarrhea**	
*Pteris cretica* L. TJ-64	Thandi booti	Wild	H	LE, FD	PA	External	**Wound**	1■, 2■, 3•, 4•, 5•, 6•, 7•, 8•, 9•, 10•, 11•, 12•, 13•, 14•, 15•, 16•, 17•, 18•, 19•, 20•, 21•, 22•
*Pteris vittata* L. TJ-34	Nanore	Wild	H	RH, LE	PA	External	**Glandular swelling**	1•, 2•, 3•, 4•, 5•, 6•, 7•, 8•, 9•, 10•, 11•, 12•, 13•, 14•, 15•, 16•, 17•, 18•, 19•, 20•, 21•, 22•
				WP	AEX	Internal	Antibacterial, Antifungal	
**Pulmbaginaceae**								
*Plumbago zeylanica* L. TJ-17	Chitra	Wild	S	WP	AEX	Internal	Stimulant, Digestant, Laxative, Muscular pain, **Rheumatic pain**	1•, 2•, 3•, 4•, 5•, 6•, 7•, 8•, 9•, 10•, 11•, 12∆, 13•, 14•, 15•, 16•, 17•, 18•, 19•, 20•, 21•, 22•
**Ranunculaceae**								
*Clematis grata* Wall. TJ-126	Tootal	Wild	C	LE	AEX	External	**Germicide ***	1•, 2•, 3•, 4•, 5•, 6■, 7•, 8•, 9∆, 10•, 11•, 12•, 13•, 14•, 15•, 16•, 17•, 18•, 19•, 20•, 21•, 22•
				LE	PA	External	Wounds	
*Clematis orientalis* L. TJ-124	Bail	Wild	C	WP	DE	Internal	**Febrifuge**	1•, 2•, 3•, 4•, 5•, 6•, 7•, 8•, 9∆, 10•, 11•, 12•, 13•, 14•, 15•, 16•, 17•, 18•, 19•, 20•, 21•, 22•
				WP	IN	Internal	Ulcerated throat	
*Thalictrum foliolosum* DC. TJ-23	Marcir/Mameera	Wild	H	RT	JU	Internal	**Stomachache**	1•, 2•, 3•, 4∆, 5•, 6•, 7■, 8•, 9•, 10•, 11•, 12•, 13•, 14•, 15•, 16•, 17•, 18•, 19•, 20•, 21•, 22•
				RT	AEX	Internal	Dysentery	
**Rhamnaceae**								
*Ziziphus jujuba* Mill. TJ-58	Beri	Wild	S	LE	CH	Internal	Lower blood glucose level *	1•, 2•, 3•, 4•, 5■, 6•, 7•, 8∆, 9•, 10•, 11•, 12•, 13•, 14•, 15•, 16∆, 17∆, 18•, 19•, 20•, 21•, 22•
				LE	CH	Internal	Skin infection	
				FR	PD	Internal	**Diabetes**	
*Ziziphus oxyphylla* Edgew. TJ-107	Beri	Wild	S	RT	AEX	Internal	**Jaundice**	1•, 2•, 3•, 4∆, 5■, 6•, 7•, 8•, 9•, 10•, 11•, 12•, 13•, 14•, 15•, 16•, 17•, 18∆, 19•, 20•, 21•, 22∆
				FR	PD	Internal	Hepatitis	
				LE	DE	Internal	Burning	
				RT, BA	IN	Internal	Hypertension	
**Rosaceae**								
*Duchesnea indica* (Jacks.) Focke Teschem. TJ-15	Budmeva	Cultivated	H	SD, LE	JU	Internal	Fever *, Jaundice *	1•, 2∆, 3•, 4•, 5∆, 6∆, 7•, 8•, 9•, 10•, 11•, 12•, 13•, 14•, 15•, 16•, 17∆, 18•, 19•, 20•, 21•, 22∆
				FR	ET	Internal	**Stomach disorders** *	
*Eriobotrya japonica* (Thunb.) Lindl. TJ-60	Locat	Cultivated	T	FR	ET	Internal	Sedative, **Vomiting**	1■, 2■, 3•, 4•, 5•, 6•, 7•, 8•, 9•, 10•, 11•, 12•, 13•, 14•, 15•, 16•, 17•, 18•, 19•, 20•, 21•, 22•
				LE	PO	External	Swelling	
*Fragaria nubicola* (Lindl. Ex Hook. f.) Lacaita TJ-55	Budi meva	Cultivated	H	FR	CH	Internal	Laxative, **Mouth infection**	1∆, 2■, 3•, 4■, 5■, 6•, 7∆, 8•, 9•, 10•, 11•, 12•, 13•, 14•, 15∆, 16•, 17•, 18•, 19•, 20•, 21•, 22•
				LE, RT	PD	Internal	Skin infection, Diarrhea	
*Prinsepia utilis* Royle TJ-06	Bekhal	Wild	S	SD	Oil	External	Rheumatism, **Muscular pain**	1•, 2•, 3•, 4•, 5•, 6•, 7•, 8•, 9•, 10•, 11•, 12•, 13•, 14•, 15•. 16•, 17•, 18•, 19•, 20•, 21•, 22•
				SD	PO	External	Treat ringworm, Eczema	
*Prunus domestica* L. TJ-49	Alubukhara	Cultivated	T	FR	JU	Internal	Jaundice, **Diabetes** *	1∆, 2∆, 3■, 4•, 5•, 6•, 7•, 8•, 9•, 10•, 11•, 12•, 13•, 14•, 15•, 16•, 17•, 18•, 19•, 20•, 21•, 22•
*Prunus persica* (L.) Batsch TJ-120	Aruu	Cultivated	T	LE	JU	Internal	Whooping, Bronchitis, **Kill intestinal worms**, Cough	1∆, 2■, 3•, 4•, 5•, 6■, 7•, 8•, 9•, 10•, 11•, 12•, 13•, 14•, 15•, 16•, 17•, 18∆, 19•, 20•, 21•, 22•
				FR	ET	Internal	Control cholesterol level, Healthy vision, Healthy teeth and bones	
*Pyrus pashia* Buch.-Ham. ex D. Don TJ-51	Tangi	Wild	T	LE	AEX	External	Tonic for hair loss *	1∆, 2∆, 3•, 4•, 5•, 6•, 7•, 8•, 9•, 10•, 11•, 12•, 13•, 14•, 15∆, 16•, 17•, 18•, 19•, 20•, 21•, 22•
				FR	ET	Internal	**Constipation**	
				LE	DE	Internal	Dysentery *, Diarrhea	
				FR	RF	Internal	Eye dark circles	
*Rosa brunonii* Lindl. TJ-68	Tarnari	Wild	S	FL	DE	Internal	Constipation	1■, 2■, 3•, 4∆, 5•, 6∆, 7∆, 8•, 9•, 10•, 11•, 12•, 13•, 14•, 15•, 16•, 17•, 18∆, 19•, 20•, 21•, 22•
				FL	PD	Internal	Diarrhea, Heart tonic, **Eye diseases**	
				LE	JU	External	Cuts, wounds	
				RT	PA	External	Scabies	
*Rubus ellipticus* Sm./TJ-07	Akhrah	Wild	S	ST	Oil	External	**Teeth pain**	1•, 2•, 3•, 4•, 5•, 6•, 7•, 8•, 9•, 10∆, 11•, 12•, 13•, 14•, 15•, 16•, 17•, 18∆, 19•, 20•, 21•, 22•
				FR	ET	Internal	Laxative	
*Rubus fruticosus* Hook. f. TJ-70	Bari. Black berry	Wild	S	LE	IN	Internal	Diarrhea, Antipyretic	1■, 2■, 3•, 4•, 5•, 6∆, 7•, 8•, 9∆, 10•, 11•, 12•, 13•, 14•, 15•, 16•, 17•, 18•, 19•, 20•, 21∆, 22•
				BA	SO	Internal	**Diabetes**	
				FR	AEX	Internal	Tonic	
*Rubus niveus* Thunb. TJ-141	Akhrah	Wild	S	LE	PD	Internal	**Diarrhea**, Fever	1■, 2■, 3•, 4•, 5•, 6•, 7•, 8•, 9•, 10•, 11•, 12•, 13•, 14•, 15•, 16•, 17•, 18•, 19•, 20•, 21•22•
				RT	DE	Internal	Dysentery, Whooping, Cough	
**Rubiaceae**								
*Galium aparine* L. TJ-61	Lainda	Wild	C	LE	PA	External	**Wound healing ***	1∆, 2∆, 3•, 4•, 5•, 6■, 7•, 8•, 9•, 10•, 11•, 12•, 13•, 14•, 15∆, 16•, 17•, 18•, 19•, 20•, 21•, 22•
				RT	JU	Internal	Fever *	
				WP	JU	Internal	Diuretic, Cancer, Urinary bladder and kidney infection	
*Rubia cordifolia* L. TJ-14	Lahndarsa bail	Wild	C	WP	PA	External	**Pimple ***, Itching *	1•, 2•, 3•, 4•, 5•, 6■, 7•, 8•, 9•, 10•, 11•, 12•, 13•, 14•, 15•, 16•, 17•, 18∆, 19•, 20•, 21•, 22•
				WP	JU	Internal	Amenorrhea, Menstruation and febrifuge	
**Rutaceae**								
*Zanthoxylum alatum* Roxb. TJ-38	Timber	Wild	S	SD	PD	Internal	Gastric, **Piles**	1•, 2•, 3•, 4∆, 5•, 6•, 7•, 8•, 9•, 10•, 11•, 12•, 13•, 14•, 15•, 16•, 17•, 18■, 19•, 20•, 21•, 22•
				FR, BR	JU	Internal	Stomach disorder, Indigestion, Piles	
**Salicaceae**								
*Populus alba* L. TJ-67	Sufaida	Wild	T	ST, BA	PD	Internal	Anti-inflammatory, antiseptic, **Diuretic**, Astringent, Tonic	1•, 2•, 3•, 4•, 5•, 6•, 7•, 8•, 9•, 10•, 11∆, 12•, 13•, 14•, 15•, 16•, 17•, 18•, 19•, 20•, 21•, 22•
				BA	PA	External	Infected wounds, Sprain	
				BA	AEX	Internal	Hemorrhoids	
*Salix nigra* Marshall TJ-121	Beesa	Wild	T	ST, BA, RT	BO	Internal	**Fever**, Headache, Paralysis	1•, 2•, 3•, 4•, 5•, 6•, 7•, 8•, 9•, 10•, 11•, 12•, 13•, 14•, 15•, 16•, 17•, 18•, 19•, 20•, 21•, 22•
				LE, BR	PA	External	Itching, Allergy	
**Sapindaceae**								
*Acer caesium* Wall. ex Brandis TJ-46	Shrub	Wild	S	WP	DE	Internal	Cardiovascular diseases, Antitumor, Antimicrobial, Anti-inflammatory	1•, 2•, 3•, 4•, 5•, 6•, 7•, 8•, 9•, 10•, 11•, 12•, 13•, 14•, 15•, 16•, 17•, 18•, 19•, 20•, 21•, 22•
				BA	JU	Internal	Eye diseases, **Bruises**	
*Aesculus indica* (Wall. ex Cambess.) Hook. TJ-112	Banakhor	Wild	T	FR	ET	Internal	Colic, Rheumatic pain	1■, 2■, 3•, 4•, 5•, 6•, 7•, 8•, 9•, 10•, 11•, 12•, 13•, 14•, 15∆, 16•, 17•, 18•, 19•, 20•, 21•, 22•
				SD	PD	Internal	**Leucorrhea**	
*Dodonaea viscosa* (L.) Jacq. TJ-08	Sanatha	Wild	S	ST	Oil	External	Teeth pain	1•, 2•, 3•, 4•, 5•, 6■, 7•, 8∆, 9•, 10•, 11•, 12•, 13•, 14•, 15•, 16•, 17•, 18■, 19•, 20•, 21•, 22•
				BA	PA	External	**Bone fracture ***	
*Sapindus mukorossi* Gaertn. TJ-40	Rantha	Cultivated	T	FR	AEX	Internal	Piles	1∆, 2∆, 3•, 4■, 5•, 6•, 7•, 8•, 9•, 10•, 11•, 12•, 13•, 14•, 15•, 16•, 17•, 18■, 19•, 20•, 21•, 22•
				FR, SD	AEX	External	**Scorpion sting and snake bite**	
				LE, RT	PA			
**Scrophulariaceae**								
*Buddleja asiatica* Lour. TJ-26	Bana	Wild	S	RT, LE	DE	Internal	Head tumor	1•, 2•, 3•, 4•, 5•, 6•, 7•, 8•, 9•, 10•, 11•, 12•, 13•, 14•, 15•, 16•, 17•, 18•, 19•, 20•, 21•, 22•
				WP	AEX	Internal	**Skin diseases**, Abortion, Loss of weight	
*Verbascum thapsus* L. TJ-127	Gidar tobacco	Wild	H	LE	DE	Internal	Diarrhea, **Dysentery**	1∆, 2∆, 3•, 4∆, 5■, 6∆, 7•, 8•, 9•, 10∆, 11•, 12•, 13•, 14•, 15∆, 16•, 17•, 18∆, 19•, 20∆, 21∆, 22•
				FP	DE	External	Skin infection	
**Simaroubaceae**								
*Ailanthus altissima* (Mill.) Swingle TJ-39	Drave	Wild	T	BA	IN	Internal	**Dysentery**, Diarrhea	1•, 2•, 3•, 4•, 5•, 6•, 7•, 8∆, 9•, 10•, 11•, 12•, 13•, 14•, 15•, 16•, 17•, 18•, 19•, 20•, 21•, 22•
**Smilacaceae**								
*Smilax aspera* L. TJ-81	Shee Bail	Wild	C	RT	PD	Internal	Depurative, Diaphoretic, **Tonic**, Diuretic	1•, 2•, 3•, 4•, 5•, 6•, 7•, 8•, 9•, 10•, 11•, 12•, 13•, 14•, 15•, 16•, 17•, 18•, 19•, 20•, 21•, 22•
**Solanaceae**								
*Physalis divaricata* D. Don TJ-20	Hundusi	Wild	H	LE	AEX	External	Wound healing, **Foot and heel cracks**	1•, 2•, 3•, 4•, 5•, 6■, 7•, 8•, 9•, 10•, 11•, 12•, 13•, 14•, 15•, 16•, 17•, 18•, 19•, 20•, 21•, 22•
				FR	PD	Internal	Diuretic, Tonic	
*Solanum nigrum* L. TJ-73	Kach Mach	Wild	H	LE	JU	Internal	**Mouth ulcer**	1•, 2•, 3∆, 4∆, 5•, 6∆, 7•, 8•, 9•, 10•, 11•, 12•, 13•, 14∆, 15•, 16∆, 17•, 18∆, 19∆, 20•, 21∆, 22∆
				SH	PD	Internal	Dropsy, Jaundice	
				FR, LE	DE	Internal	Swelling of body, Cough	
**Taxaceae**								
*Taxus wallichiana* Zucc. TJ-77		Wild	T	FR, LE	IN	Internal	Whooping, Bronchitis, Asthma, **Cough**	1•, 2•, 3•, 4•, 5•, 6•, 7•, 8■, 9∆, 10∆, 11•, 12•, 13•, 14•, 15•, 16•, 17•, 18•, 19•, 20•, 21•, 22•
**Violaceae**								
*Viola canescens* Wall. TJ-25	Gulbanafsha	Wild	H	FL, LE	JU	Internal	**Fever**, Cough, Throat irritation * Nervous disorders, Laxative	1■, 2■, 3•, 4•, 5•, 6■, 7•, 8∆, 9•, 10•, 11•, 12•, 13•, 14•, 15•, 16•, 17•, 18∆, 19•, 20•, 21∆, 22•

Acronyms: H, Herb; S, Shrub, T, Tree; C, Climber; LE, Leaf; FR, Fruit; RT, Root; ST, Stem; AP, Aerial Parts; WP, Whole Plant; FD, Fronds; SD, Seed; FL, Flower; BA, Bark; BL, Bulb; RH, Rhizome; SH, Shoot; WO, Wood Oil; BR, Branches; FP, Floral parts; RS, Resin; TU, Tuber; CO, Corm; VP, Vegetable Part 2. Method of Preparation: PD, Powder; DE, Decoction; AEX, Aqueous Extract; PA, Paste; JU, Juice; PO, Poultice; IN, Infusion; CH, Chewed; TE, Tea; RB, Rubbing; ET, Eaten; CK, Cooked; BO, Boiled; SO, Soaked; RF, Raw form; LX, Latex. (■) = Similar use, (∆) = Dissimilar use, (•) = Use not reported, (*) = Use not reported in previous study. Uses in **bold** highlight the specific and preferred uses of the respective plant. 1 *=* Shaheen et al. (2017) [44]; 2 *=* Amjad et al. (2017) [47]; 3 *=* Bibi et al. (2014) [46]; 4 *=* Khan et al. (2012) [72]; 5 *=* Ahmad et al. (2014) [45]; 6 *=* Amjad et al. (2017) [23]; 7 *=* Kayani et al. (2015) [49]; 8 *=* Jan et al. (2011) [73]; 9 *=* Ishtiaq et al. (2012) [74]; 10 *=* Ahmad et al. (2017) [75]; 11 *=* Sargin et al. (2013) [76]; 12 *=* Musa et al. (2011) [77]; 13 *=* Hada and Katewa (2015) [78]; 14 *=* Umair et al. (2017) [79]; 15 *=* Rahman et al. (2016) [80]; 16 *=* Noreen et al. (2018) [81]; 17 *=* Aziz et al.(2018) [31]; 18 *=* Qaseem et al. (2019) [82]; 19 *=* Umair et al. (2019) [83]; 20 *=* Khan et al. (2015) [84]; 21 *=* Hussain et al. (2018) [85]; 22 *=* Khan et al. (2013) [86].

## Data Availability

The original contributions presented in the study are included in the article; further inquiries can be directed to the corresponding authors.

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
