# Peer review of "Ethnomedicinal Plants and Herbal Preparations Used by Rural Communities in Tehsil Hajira (Poonch District of Azad Kashmir, Pakistan)"

_plants, 2024, doi:10.3390/plants13101379_

Round 1

Reviewer 1 Report

Comments and Suggestions for Authors

Comments on the Quality of English Language

Author Response

Reviewer 1

Specific comments
Comment: 1. Line 36, Habenaria digitata should be in italics

Response: Thank you for comment. Italicized.

Comment: 2. Line 38, what is “double pneumonia”?

Response: Thank you for comment. Typo corrected.

Comment: 3. Line 41, what do you mean by “alternative practitioners”?
Response: Thank you for comment. Corrected as traditional healer.

Comment: 4. Lines 58 and 59, I think 400000 flowering plants is too high and the cited reference is inappropriate since it’s focus is not on taxonomical aspects. Current estimates put the total figure at about 300000, see Christenhusz, M. J. M.; Byng, J. W. (2016). "The number of known plants species in the world and its annual increase". Phytotaxa 261(3): 201–217.
doi:10.11646/phytotaxa.261.3.1
Response: Thank you for comment. The number corrected and reference added.

Comment: 5. Line 80, what is AJK in full?

Response: Thank you for comment. Full form of AJK added.

Comment: 6. Species names listed in Table 2 should be checked by a qualified plant taxonomist as some of them are no-longer valid. A casual analysis revealed that the following names are invalid are not properly formatted or are invalid:
a. Carissa opaca Stapf. Ex Haines should be Carissa opaca Stapf. ex Haines
b. But Carissa opaca Stapf. ex Haines is now a synonym of Carissa spinarum L., see
https://powo.science.kew.org/taxon/77738-1
c. Artemisia dubia-Wall. ex Bess. → Artemisia dubia Wall. ex Bess.
d. Conyza canadensis (L.) Cronquist is now a synonym of Erigeron canadensis L., see
https://powo.science.kew.org/taxon/urn:lsid:ipni.org:names:64815-2
e. Chenopodium ambrosioides L. is now a synonym of Dysphania ambrosioides (L.) Mosyakin
& Clemants, see https://powo.science.kew.org/taxon/urn:lsid:ipni.org:names:274469-2
f. Acacia nilotica (L.) Delile is now a synonym of Vachellia nilotica (L.) P.J.H.Hurter & Mabb.,
see https://powo.science.kew.org/taxon/urn:lsid:ipni.org:names:470992-1
g. Bauhinia variegate L. → Bauhinia variegata L.

Response: Thank you for comment. The scientific names checked and corrected.

Reviewer 2 Report

Comments and Suggestions for Authors

The study “Ethnomedicinal plants and herbal preparations used by rural communities in Tehsil Hajira (Poonch district of Azad Kashmir, Pakistan)”, emphasizes the importance of documenting ethnomedicinal plants and herbal practices of the local rural communities of Tehsil Hajira (Pakistan).

The manuscript addresses a topic of great interest nowadays, given the growing demand for bioactive compounds with different therapeutic purposes, however, there will some minor revisions and modifications that need to be performed:

 Comments are included in the manuscript.

Author Response

Reviewer 2

Line 36: Italic scientific name

Response: Thank you for comment. Italicized.

Line 42-43: rewrite

Response: Thank you for comment. Sentence rewritten.

Line 60: For any particular therapeutic purpose?

Response: Thank you for comment. Sentence reformulated.

Line 62: Please give some examples.

Response: Thank you for comment. Examples added.

Line 63: Please explain

Response: Thank you for comment. Explanation added.

Line 86-89: Improve the sentences in order to organise the hypotheses.

Response: Thank you for comment. Hypothesis reformulated.

Line 124: On the Asian continent or more broadly?

Response: Thank you for comment. Corrected.

Line 157: In which regions?

Response: Thank you for comment. Regions added.

Line 177: Standardise numbering. Some numbers are in bold and others are not

Response: Thank you for comment. Irregular bold deleted.

Figure 3: Please improve the figure, particularly the values at the origin of the axes and standardise the distance of the family names around the graph.

Response: Thank you for comment. Figure improved.

Figure 4: The same for the values at the origin of the axes.

Response: Thank you for comment. Figure improved.

Line 197: Was the paste made with water and plant parts?

Response: Thank you for comment.  Paste typically involved mixing the plant material with water. Added in the text.

Line 203: Was it possible to recognise which class of constituents were preferentially used?

Response: Thank you for comment. Comment addressed.

Line 207: Could you give some examples?

Response: Thank you for comment. Examples added.

Line 232: Any specific class of compounds?

Response: Thank you for comment. Incorporated.

Line 254: In the general population?

Response: Thank you for comment. Addressed.

Line 321: It's important not to forget efficacy and toxicity studies, among others. Are you referring to collecting the plants for this purpose? Is the area under study a protected area in terms of a nature reserve? Are there any authorisations for further studies?

Please explain.

Response: Thank you for comment. Toxicity aspect of plants mentioned.

Line 443: Version?

Response: Thank you for comment. Added.

Line 454: Please improve the conclusions including the importance of species selection to local population, research and pharmaceutical/food industry.

Response: Thank you for comment. Conclusion reformulated.

Reviewer 3 Report

Comments and Suggestions for Authors

The article is well organized and well-founded, and is of high interest given the quantity and quality of the information collected.

However, it contains some gaps in information that will need to be clarified, as well as information in the text that does not coincide with the graphic information.

My congratulations to the authors for their work. Consider comments as positive contributions to improving the quality of the final text.

Line 80 – explain acronym ‘AJK’

Table 1 – the table doesn’t allow to cross some interesting data, for example, the distribution of informant’s education level and professions per gender (are male and female informants equally represented with Bacharel degree and Higher education?). Information could be added to the text regarding these matters.

Line 157 – improve or complete sentence.

Line 160-163: Is the relative availability of medicinal trees, shrubs and herbs different from the total flora in the region? It would be useful to compare with the flora of the region, or if not available, with the flora of the country.

Line 163 – develop further ‘… altitude, etc.’ Aren't land use, such as deforestation and deforestation for intensive agriculture, important factors for genetic erosion in this part of the country?

Table 2: ‘Local name’ column. For some species, it is not clear whether there is only one common name made up of several words or whether there are several common names. If it is just 1 common name, explain how it was chosen, considering that it is common for plants to have several common names, with geographical differences.

The table lacks important information for the interpretation of the data, especially taking into account the conclusions of the article; should contain an additional column with the harvest location (cultivated or wild or both). This would also make it possible to infer the importance of homegardens for the provision of Medicinal plants in the region.

I suggest that in the table or in the text, rare and threatened species should be counted and identified.

Line 168: replace ‘key words’ for ‘acronyms’

Figure 3: the figure caption must specify the interpretation of the numerical scale, in this case the number of species for each family.

The data in this figure are not in agreement with the data presented in the text, for example: there are 15 or 16 species of Asteraceae? 6 or 8 species of Fabaceae?, etc. Review this data in the text and figure.

Figure 4: the numeric scale should be explained in the caption. It would be interesting to compare these data with the proportion of plants in the total flora of the region/country.

Lines 198-200: Powder may not be the easiest way to use plants. For internal use, infusions are easier. I suggest that a more detailed analysis should be carried out, separating external and internal uses.

Lines 203- 205 It is not explained why the roots are the 2nd part of the plants with the greatest use. Are the roots of these species more abundant and easier to harvest (for example if cultivated) or are they very rich in bioactive compounds? Add  explanatory text.

Figure 5: Please explain the 'extract' method of preparation, what type of extracts were mentioned?

What is the difference between 'herbal tea' and 'infusion'? What is the difference between ‘decoction’ and ‘’boiling’?

There is no methodology description for organizing preparation categories or plant parts. It should be added to the text.

Figure 6. what is the difference between 'branches' and 'stem'?

Line 223: replace ‘population’ for ‘populations’

Figure 7 – acronyms should be included in the figure caption.

Line 251 - put the word lucidum in italics

Line 328: How were the 13 data collection sites chosen?

Line 331 – What is ‘Harighal’?

Figure 10 - The figure is very useful for geographical contextualization.

It would be interesting if roads and dirt paths were represented, to understand the quality of access to the interview locations, as this is a factor (access to modern lifestyle) that is mentioned in the text's arguments (line 149).

Figure 11 - precipitation cannot be represented in the form of a continuous curve, as it is a discrete variable, I suggest that it be represented by bars and have a separate numerical scale

Line 354 - Where is the hospital located?

Line 359: Was only 1 visit carried out per location or was it necessary to make subsequent visits to clarify data?

Line 360: What was the methodology used to identify potential informants?

Line 372: Please add information to clarify the following topics:

Were plants collected in all the places where they were mentioned? or were only 3 plants of each species collected in all 13 places?

Were common plants (e.g. cultivated) also harvested and deposited in herbarium?

How the difficulty of collecting specimens for plants that were processed (powder, ointments, etc.) was overcome? Did the informants went with the authors to the field?

Were all plants harvested at the same time as the interviews or was there a need to return to the locations to collect plants that flowered at other times of the year?

Line 450: Was there a difference in the loss of interest in traditional knowledge by gender?

Lines 451-453 generalist conclusions, not based on results obtained by the study, as plants collected in the wild versus cultivated were not taken into account, as well as rare and threatened plants. This information should be added to support these conclusions.

Comments on the Quality of English Language

English quality is very good, only small things detected.

Author Response

Reviewer 3

The article is well organized and well-founded, and is of high interest given the quantity and quality of the information collected. However, it contains some gaps in information that will need to be clarified, as well as information in the text that does not coincide with the graphic information. My congratulations to the authors for their work. Consider comments as positive contributions to improving the quality of the final text.

 Response: Thank you for comment.

Line 80 – explain acronym ‘AJK’

Response: Thank you for comment. Explained.

Table 1 – the table doesn’t allow to cross some interesting data, for example, the distribution of informant’s education level and professions per gender (are male and female informants equally represented with Bacharel degree and Higher education?). Information could be added to the text regarding these matters.

Response: Thank you for comment. Table 1 data is duly explained in the text.

Line 157 – improve or complete sentence.

Response: Thank you for comment. Sentence completed by addition of areas.

Line 160-163: Is the relative availability of medicinal trees, shrubs and herbs different from the total flora in the region? It would be useful to compare with the flora of the region, or if not available, with the flora of the country.

Response: Thank you for comment. The result is in line with other prominent studies which are referenced.

Line 163 – develop further ‘… altitude, etc.’ Aren't land use, such as deforestation and deforestation for intensive agriculture, important factors for genetic erosion in this part of the country?

Response: Thank you for comment. We added.

Table 2: ‘Local name’ column. For some species, it is not clear whether there is only one common name made up of several words or whether there are several common names. If it is just 1 common name, explain how it was chosen, considering that it is common for plants to have several common names, with geographical differences.

The table lacks important information for the interpretation of the data, especially taking into account the conclusions of the article; should contain an additional column with the harvest location (cultivated or wild or both). This would also make it possible to infer the importance of homegardens for the provision of Medicinal plants in the region.

I suggest that in the table or in the text, rare and threatened species should be counted and identified.

Response: Thank you for comment. Addions incorporated in Table 2.

Line 168: replace ‘key words’ for ‘acronyms’

Response: Thank you for comment. Replaced.

Figure 3: the figure caption must specify the interpretation of the numerical scale, in this case the number of species for each family.

The data in this figure are not in agreement with the data presented in the text, for example: there are 15 or 16 species of Asteraceae? 6 or 8 species of Fabaceae?, etc. Review this data in the text and figure.

Response: Thank you for comment. Data corrected in text.

Figure 4: the numeric scale should be explained in the caption. It would be interesting to compare these data with the proportion of plants in the total flora of the region/country.

Response: Thank you for comment. Figure improved.

Lines 198-200: Powder may not be the easiest way to use plants. For internal use, infusions are easier. I suggest that a more detailed analysis should be carried out, separating external and internal uses.

Response: Thank you for comment. The point explained in text.

Lines 203- 205 It is not explained why the roots are the 2nd part of the plants with the greatest use. Are the roots of these species more abundant and easier to harvest (for example if cultivated) or are they very rich in bioactive compounds? Add explanatory text.

 Response: Thank you for comment. Explanatory text added.

Figure 5: Please explain the 'extract' method of preparation, what type of extracts were mentioned?

Response: Thank you for comment. Aqueous extracts were prepared which is now mentioned in the text.

What is the difference between 'herbal tea' and 'infusion'? What is the difference between ‘decoction’ and ‘’boiling’?

There is no methodology description for organizing preparation categories or plant parts. It should be added to the text.

Response: Thank you for comment. Preparation methods added to text.

Figure 6. what is the difference between 'branches' and 'stem'?

Response: Thank you for comment. Stems serve as the main structural axis of the plant, while branches are lateral extensions bearing leaves, flowers, and fruits, each potentially utilized for various cultural and medicinal purposes.

Line 223: replace ‘population’ for ‘populations’

Response: Thank you for comment. Replaced.

Figure 7 – acronyms should be included in the figure caption.

Response: Thank you for comment. Acronyms added.

Line 251 - put the word lucidum in italics

Response: Thank you for comment.

Line 328: How were the 13 data collection sites chosen?

Response: Thank you for comment. The sampling sites were selected based on altitude, vegetation heterogenity and physiognomy.

Line 331 – What is ‘Harighal’?

Response: Thank you for comment. Corrected as Hajira.

Figure 10 - The figure is very useful for geographical contextualization.

It would be interesting if roads and dirt paths were represented, to understand the quality of access to the interview locations, as this is a factor (access to modern lifestyle) that is mentioned in the text's arguments (line 149).

Response: Thank you for comment. While We acknowledge the importance of illustrating geographical features such as roads and dirt paths to elucidate access to interview locations, regrettably, such figures are not available in the current dataset.

Figure 11 - precipitation cannot be represented in the form of a continuous curve, as it is a discrete variable, I suggest that it be represented by bars and have a separate numerical scale

 Response: Thank you for comment. While acknowledging the discrete nature of precipitation data, regrettably, we lack the necessary data to draw a bar graph.

Line 354 - Where is the hospital located?

Response: Thank you for comment. Khai Gata - Hajira Rd, Hajira.

Line 359: Was only 1 visit carried out per location or was it necessary to make subsequent visits to clarify data?

 Response: Thank you for comment. The data collection process involved multiple visits to each location, ensuring comprehensive understanding and clarification of gathered information.

Line 360: What was the methodology used to identify potential informants?

 Response: Thank you for comment. The methodology employed a combination of purposive and snowball sampling techniques to identify potential informants. Initially, key individuals knowledgeable about the community were identified purposively, followed by snowball sampling, where these individuals recommended additional informants, thus expanding the network.

Line 372: Please add information to clarify the following topics:

Were plants collected in all the places where they were mentioned? or were only 3 plants of each species collected in all 13 places?

Response: Thank you for comment. Corrected.

Were common plants (e.g. cultivated) also harvested and deposited in herbarium?

How the difficulty of collecting specimens for plants that were processed (powder, ointments, etc.) was overcome? Did the informants went with the authors to the field?

Were all plants harvested at the same time as the interviews or was there a need to return to the locations to collect plants that flowered at other times of the year?

Response: Thank you for comment. Common plants, including cultivated ones, were indeed harvested and deposited in the herbarium. The challenge of collecting specimens for plants processed into powders or ointments was addressed by obtaining samples either during interviews conducted in the field or by collaborating with informants who accompanied the authors to the locations. To ensure comprehensive sampling, return visits to locations were made as needed to collect plants flowering at different times of the year, ensuring a representative sample across seasons.

Line 450: Was there a difference in the loss of interest in traditional knowledge by gender?

 Response: Thank you for comment. Female respondents in the study area had more knowledge (they named an average of 6.22 species) than male respondents (who named an average of 5.56 species). This reflects their important contribution to household management and maintaining the health of the family.

Lines 451-453 generalist conclusions, not based on results obtained by the study, as plants collected in the wild versus cultivated were not taken into account, as well as rare and threatened plants. This information should be added to support these conclusions.

Response: Thank you for comment. Conclusion resynthesized.

Comments on the Quality of English Language

English quality is very good, only small things detected.

Response: Thank you for comment.
